# BAYES ADAPTIVE MONTE CARLO TREE SEARCH FOR OFFLINE MODEL-BASED REINFORCEMENT LEARNING

## ABSTRACT

Offline reinforcement learning (RL) is a powerful approach for data-driven decision-making and control. Compared to model-free methods, offline model-based reinforcement learning (MBRL) explicitly learns world models from a static dataset and uses them as surrogate simulators, improving the data efficiency and enabling the learned policy to potentially generalize beyond the dataset support. However, there could be various MDPs that behave identically on the offline dataset and so dealing with the uncertainty about the true MDP can be challenging. In this paper, we propose modeling offline MBRL as a Bayes Adaptive Markov Decision Process (BAMDP), which is a principled framework for addressing model uncertainty. We further introduce a novel Bayes Adaptive Monte-Carlo planning algorithm capable of solving BAMDPs in continuous state and action spaces with stochastic transitions. This planning process is based on Monte Carlo Tree Search and can be integrated into offline MBRL as a policy improvement operator in policy iteration. Our "RL + Search" framework follows in the footsteps of superhuman AIs like AlphaZero, improving on current offline MBRL methods by incorporating more computation input. The proposed algorithm significantly outperforms state-of-the-art model-based and model-free offline RL methods on twelve D4RL MuJoCo benchmark tasks and three target tracking tasks in a challenging, stochastic tokamak control simulator.

## 1 INTRODUCTION

The success of reinforcement learning (RL) typically relies on large amounts of interactions with the environment. However, in real-world scenarios, such interactions can be unsafe or costly. As an alternative, offline RL (Levine et al. (2020)) leverages offline datasets of transitions, collected by a behavior policy, to train a policy that can transfer to an online task. To avoid overestimation of the value function for some (out-of-sample) states in the environment, which can mislead policy learning, model-free offline RL methods (Kumar et al. (2020); Wu et al. (2019)) often constrain the learned policy to remain close to the behavior policy or within the support of the offline dataset. However, collecting transitions that comprehensively cover possible task scenarios, or acquiring a large volume of demonstrations from a high-quality behavior policy, can be expensive. This challenge has led to the development of offline model-based reinforcement learning (MBRL) approaches, such as (Lu et al. (2022); Guo et al. (2022)). These methods train dynamics models from offline data and optimize policies using imaginary rollouts generated by the models. Notably, the dynamics modeling is independent of the behavior policy, making it possible to achieve high returns even with data collected from a random policy. Furthermore, with careful dynamics modeling and thorough simulation, the learned policy can more effectively handle the environmental stochasticity and generalize to states beyond the support of the offline dataset.

Given a dataset, there may be various potential MDPs that behave identically on the limited set of states and actions, but their dynamics and reward functions could differ, especially on out-of-sample states and actions. This implies that we are dealing with a distribution of possible world models underlying the dataset. A common strategy in offline MBRL is to learn an ensemble of world models and treat them equally. For instance, when determining the next state, a world model is uniformly sampled from the ensemble and generate its prediction. However, different ensemble members may perform better in different regions of the state-action space, making it necessary to adapt the belief over each ensemble member based on the experience accumulated since the start

of the episode. The Bayes Adaptive Markov Decision Process (BAMDP, Duff (2002)) provides a principled framework for modelling such an adaptive process. We show in Section 4.1 that, despite the need for Bayesian posterior updates, BAMDPs can still be efficiently simulated using deep ensembles. BAMCP (Guez et al. (2013)) is an efficient online planning method for solving BAMDPs. However, BAMCP has several limitations: (1) it relies on a ground-truth world model for planning; (2) it is restricted to discrete state and action spaces; and (3) its outcome is an action choice at a particular state, rather than a policy function. To address these challenges: (1) we apply a reward penalty (defined with the adapting belief) to construct a pessimistic BAMDP, preventing overexploitation of inaccurate world models (learned from the offline dataset); (2) we propose a novel planning algorithm to solve BAMDPs in continuous state and action spaces by extending BAMCP with double progressive widening (Auger et al. (2013)); and (3) we integrate the planning component as a policy improvement operator within policy iteration RL methods (Sutton & Barto (2018)), enabling the derivation of a policy suitable for real-time execution from the planning results. Specifically, the planning process is carried out through Monte Carlo Tree Search on a BAMDP. Integrating search with RL allows for the use of significantly more computation input, thereby improving policy learning performance. Grounded in the "scaling law", this paradigm has seen tremendous success in sophisticated policy learning, as demonstrated in (Silver et al. (2017b); Schrittwieser et al. (2020); AlphaProof & AlphaGeometry (2024)). Its application to offline MBRL, particularly in continuous control tasks, is a promising area for exploration.

To summarize, the main contributions of this work include: (1) Introducing BAMDPs to handle model uncertainties in offline MBRL; (2) Proposing an efficient Bayes Adaptive Monte Carlo Tree Search method for planning in continuous, stochastic BAMDPs; (3) Developing the first algorithm to successfully integrate Bayesian RL, offline MBRL, and deep search for sophisticated policy learning in continuous control under highly stochastic environments; (4) Demonstrating the improvements brought by Bayesian RL and deep search across twelve D4RL MuJoCo tasks and three target tracking tasks in a stochastic tokamak control scenario (for nuclear fusion), highlighting the potential of our algorithm to tackle challenging, real-world problems.

## 2 BACKGROUND

A Markov Decision Process (MDP, Puterman (2014)) is described as a tuple $\mathcal{M} = \langle \mathcal{S}, \mathcal{A}, \mathcal{P}, \mathcal{R}, \gamma \rangle$. $\mathcal{S}$ and $\mathcal{A}$ are the state space and action space, respectively. $\mathcal{P} : \mathcal{S} \times \mathcal{A} \to \Delta_{\mathcal{S}}$ is the dynamics function and $\mathcal{R} : \mathcal{S} \times \mathcal{A} \to \Delta_{[0,1]}$ is the reward function, where $\Delta_{\mathcal{X}}$ denotes the set of possible probability distributions on $\mathcal{X}$. $\gamma \in [0, 1)$ is a discount factor. A Bayes Adaptive MDP (BAMDP, Duff (2002)) can model scenarios where the precise MDP $\mathcal{M}_\theta = \langle \mathcal{S}, \mathcal{A}, \mathcal{P}_\theta, \mathcal{R}_\theta, \gamma \rangle$ is uncertain but is known to follow a prior distribution $b_0(\theta)$. During planning, a Bayes-optimal agent would update its belief over the MDP based on experience. Formally, a BAMDP can be described as a tuple $\mathcal{M}^+ = \langle \mathcal{S}^+, \mathcal{A}, \mathcal{P}^+, \mathcal{R}^+, \gamma \rangle$. $\mathcal{S}^+$ denotes the space of information states $(s, b)$, which is a composition of the physical state and the current belief over the MDP. After each transition $(s, a, r, s')$, the belief is updated to the corresponding Bayesian posterior: $b'(\theta) \propto b(\theta)P((s, a, r, s')|\theta) = b(\theta)\mathcal{P}_\theta(s'|s, a)\mathcal{R}_\theta(r|s, a)$. Accordingly, $\mathcal{P}^+$ and $\mathcal{R}^+$ can be defined as follows:

$$\mathcal{P}^+((s', b'')|(s, b), a) = \mathbb{1}(b'' = b') \int_\theta \mathcal{P}_\theta(s'|s, a)b(\theta)d\theta, \ \mathcal{R}^+((s, b), a) = \int_\theta \mathcal{R}_\theta(s, a)b(\theta)d\theta \quad (1)$$

The Q-function that satisfies the Bellman optimality equations: $(\forall x = (s, b) \in \mathcal{S}^+, a \in \mathcal{A})$

$$Q^*(x, a) = \mathcal{R}^+(x, a) + \gamma \int_{x'} V^*(x')\mathcal{P}^+(x'|x, a)dx', \ V^*(x') = \max_a Q^*(x', a) \quad (2)$$

is the Bayes-optimal Q-function and $\pi^*(s, b) = \arg\max_a Q^*((s, b), a)$ is the Bayes-optimal policy. Actions derived from $\pi^*$ are executed in the real MDP and constitute the best course of action for a Bayesian agent with respect to its prior belief $b_0$ over the environment (Guez et al. (2014)). A BAMDP can be cast into a partially observable MDP (POMDP, Littman (2009)) by viewing $\mathcal{S}^+$ and $\mathcal{S}$ as the state and observation spaces, respectively. As a result, approaches developed for POMDPs can potentially be used to solve BAMDPs.

Bayesian Reinforcement Learning (BRL, Ghavamzadeh et al. (2015)), as introduced above, is a principled approach to dealing with uncertainty in the world model $\mathcal{M}_\theta$ and has two main advantages:

(1) Domain knowledge can be injected by defining a proper prior belief; (2) A Bayes Adaptive policy solves the exploration-exploitation dilemma by explicitly including the belief in its state representation and incorporating belief updates into the planning process (Sorg et al. (2010)). Bayes-optimal planning is generally intractable, and we introduce some approximate methods in the next section.

## 3 RELATED WORKS

**Offline model-based RL:** Offline RL (Chen et al. (2024)) enables an agent to learn control policies from datasets of environment transitions pre-collected by a behavior policy $\mu$, i.e., $\mathcal{D}_\mu = \{[(s_t^i, a_t^i, r_t^i)_{t=1}^T]_{i=1}^N\}$. Offline Model-based RL (MBRL) methods explicitly learn world models $\mathcal{M}_\theta$ from $\mathcal{D}_\mu$ and adopt $\mathcal{M}_\theta$ as a surrogate simulator, enabling the learned policy to possibly generalize to states beyond $\mathcal{D}_\mu$. Specifically, both planning methods (Argenson & Dulac-Arnold (2021); Zhan et al. (2022); Diehl et al. (2023)) and RL methods (Yu et al. (2020); Kidambi et al. (2020); Lu et al. (2022); Yu et al. (2021); Guo et al. (2022)) can be applied on top of the learned $\mathcal{M}_\theta$ to obtain a policy. However, since $\mathcal{D}_\mu$ may not span the entire state-action space, $\mathcal{M}_\theta$ is unlikely to be globally accurate. Learning/Planning without any safeguards against such model inaccuracy can yield poor results. In this case, the authors of (Yu et al. (2020); Kidambi et al. (2020); Lu et al. (2022)) propose learning an ensemble of world models, using ensemble-based uncertainty estimations to construct a pessimistic MDP (P-MDP), and learning a near-optimal policy atop it. Ideally, for any policy, the performance in the real environment is lower-bounded by the performance in the corresponding P-MDP (with high probability), thus avoiding being overly optimistic about an inaccurate model. Notably, none of these offline MBRL methods have modeled the problem as a BAMDP, even though Bayesian RL provides a principled framework for handling model uncertainty.

**MCTS for model-based RL**: Monte-Carlo Tree Search (MCTS, Browne et al. (2012)) has been successfully integrated with RL, as exemplified by AlphaZero (Silver et al. (2017a)) and MuZero (Schrittwieser et al. (2020)). These methods have achieved superhuman performance in domains requiring highly sophisticated decision-making processes. AlphaZero relies on given world models, whereas MuZero learns the world model and policy simultaneously by interacting with the environment. Although there have been various extensions of MuZero (Hubert et al. (2021); Schrittwieser et al. (2021); Ye et al. (2021); Antonoglou et al. (2022); Oren et al. (2022); Zhao et al. (2024)), most algorithms are designed for online MBRL. According to Niu et al. (2023), the applications of MuZero in offline learning, especially for continuous control in highly stochastic environments, which is our focus, still require significant improvement. Our algorithm design differs from MuZero in several key ways: (1) MuZero integrates model learning and policy training into a single stage, using a world model defined in a latent state space. Our algorithm separately learns a world model and then trains the policy on top of it, aligning with the widely-adopted offline MBRL framework. (2) MuZero employs a single latent model (rather than an ensemble) and does not account for uncertainty in dynamics or reward predictions. (3) We introduce double progressive widening (Auger et al. (2013)) and Bayes-adaptive planning into MCTS, making our core planning algorithm novel.

**Bayes-adaptive planning:** Bayes-optimal planning is typically intractable. Approximate methods, such as (Asmuth et al. (2009); Sorg et al. (2010); Castro & Precup (2010); Asmuth & Littman (2011); Wang et al. (2012); Fonteneau et al. (2013); Guez et al. (2013); Slade et al. (2020)) have been developed. As a representative work, BAMCP (Guez et al. (2013)) adopts MCTS for Bayes-adaptive planning and is shown to converge in probability to a near Bayes-optimal policy at the root node of the search tree. However, all these methods cannot be directly applied to large-scale MDPs with continuous state and action spaces. Moreover, these planning algorithms are not designed for offline MBRL. Thus, how to incorporate search-based planning for policy improvement in RL and how to handle the model uncertainty during planning still require exploration.

To sum up, our algorithm introduces a Bayesian approach to offline MBRL and leverages tree search to enhance policy learning. There has been related research in both directions. (1) Dorfman et al. (2021); Choshen & Tamar (2023) propose to model offline Meta RL as a BAMDP and learn a belief-conditioned policy capable of adapting to different underlying MDPs for multi-task purposes. Ghosh et al. (2022) apply the BAMDP framework to model-free offline RL, arguing that optimal policies for offline RL should be adaptive to all observed transitions. Nevertheless, these works do not explore the Bayesian treatment of model-based RL. (2) Model-based planning results can be utilized to improve the sample efficiency of model-free RL. For instance, Feinberg et al. (2018) propose Model-

based Value Expansion. It uses the learned world model to generate imaginary rollouts, providing a more accurate estimation of value function targets for online actor-critic training. This idea is later extended to the offline RL setting by Jeong et al. (2023). However, they do not employ BAMDP for uncertainty treatment, and, compared to model-based rollouts, MCTS can offer more exhaustive exploration, crucial for tackling complex tasks.

# 4 METHODOLOGY

We propose a novel offline MBRL algorithm based on Bayes Adaptive MCTS. The core challenge is to design a Bayes Adaptive planning method that is efficient in large stochastic MDPs. In this case, we propose Continuous BAMCP in Section 4.2, which can be applied to continuous control tasks with high dynamics stochasticity. Then, in Section 4.3, we present a search-based policy iteration framework, where the search results are distilled into policy and value networks for policy improvement and policy evaluation, respectively, at each iteration. In this way, we integrate offline MBRL with Bayes Adaptive MCTS. Both components require the use of an ensemble of world models for either practical implementation or uncertainty quantification, as detailed in Section 4.1.

## 4.1 THE KEY ROLE OF DEEP ENSEMBLES

Offline MBRL methods estimate world models $\mathcal{M}_\theta$ from a static dataset $\mathcal{D}_\mu$, which would inevitably induce epistemic uncertainty about the identity of the real MDP $\mathcal{M}^*$. Specifically, there could be various potential MDPs that behave identically on the limited set of states and actions in $\mathcal{D}_\mu$, but their dynamics and reward functions may differ, especially on out-of-sample states and actions. Thus, we are actually dealing with a distribution of world models that follow a prior distribution $b_0(\theta) \triangleq P(\mathcal{M}_\theta | \mathcal{D}_\mu)$. As introduced in Section 2, Bayesian RL based on BAMDP is a principled framework for handling model uncertainty by explicitly including the belief over the models in its state representation. Essentially, the belief is updated with experience, providing a measure of how the models' uncertainty has changed since the beginning of the episode. As a result, the agent can adjust its behavior upon receiving new information that reduces the epistemic uncertainty. Such an adaptive policy is necessary to act optimally in offline RL, as demonstrated in (Ghosh et al. (2022)).

The idea of deep ensembles (Lakshminarayanan et al. (2017)) is to train multiple deep neural networks as approximations of a function, each using a different weight initialization and optimized with a different mini-batch sequence. For offline MBRL, we can learn an ensemble of dynamics models $\{\mathcal{P}_\theta^1, \cdots, \mathcal{P}_\theta^K\}$ and reward models $\{\mathcal{R}_\theta^1, \cdots, \mathcal{R}_\theta^K\}$[1] from the dataset $\mathcal{D}_\mu$ by minimizing the following supervised learning loss: ($i = 1, \cdots, K$)

$$\mathcal{L}(\mathcal{P}_\theta^i) = -\mathbb{E}_{(s,a,s') \sim \mathcal{D}_\mu} \left[ \log \mathcal{P}_\theta^i(s'|s,a) \right], \ \mathcal{L}(\mathcal{R}_\theta^i) = -\mathbb{E}_{(s,a,r) \sim \mathcal{D}_\mu} \left[ \log \mathcal{R}_\theta^i(r|s,a) \right] \quad (3)$$

$\{(\mathcal{P}_\theta^i, \mathcal{R}_\theta^i)_{i=1}^K\}$ can be viewed as a set of independent and identically distributed (IID) samples from the prior $P(\mathcal{M}_\theta | \mathcal{D}_\mu)$ and constitute a finite approximation of the space of world models. With such an ensemble, the belief over the world models can be converted to a mass function over a set of $K$ items, where the $i$-th element denotes the probability of being in the MDP $(\mathcal{P}_\theta^i, \mathcal{R}_\theta^i)$. In this case, a reasonable prior distribution is $b_0(\theta) = [1/K, \cdots, 1/K]$, since these models are IID prior samples. After receiving a transition $(s, a, r, s')$, the belief can be updated as follows:

$$b'(\theta)(i) = x^i / \sum_{j=1}^K x^j, \ x^i = b(\theta)(i) \mathcal{P}_\theta^i(s'|s,a) \mathcal{R}_\theta^i(r|s,a) \quad (4)$$

This update requires a single inference from each ensemble member, but can be parallelized for computational efficiency. Equation (4) is a practical implementation of the Bayesian posterior update based on deep ensembles, where $b(\theta)$, $b'(\theta)$, and $\mathcal{P}_\theta^i(s'|s,a)\mathcal{R}_\theta^i(r|s,a)$ denote the prior, posterior distributions, and likelihood, respectively. This simplified definition of $b(\theta)$ also enables efficient execution of transitions in Bayesian RL, as described in Equation (1).

The ensemble can also be used for uncertainty quantification. As aforementioned, our algorithm relies on thorough search on the learned world models. Without any constraints on the search process,

---

[1]In some MBRL scenarios, a certain reward function is available, for instance, as defined by domain experts. Otherwise, the reward and dynamics function $(\mathcal{R}_\theta^i, \mathcal{P}_\theta^i)$ are usually trained as a unified probabilistic model $\mathcal{N}(\mu_\theta^i, \sigma_\theta^i)$, since the reward $r$ can be viewed as an element of the next state $s'$.

---

**Algorithm 1** Continuous BAMCP

**Input:** $\pi$, $V$, $E$, $d_{\max}$, $\gamma$, $\alpha$, $\beta$, $\mathcal{P}_\theta^{1:K}$, $\mathcal{R}_\theta^{1:K}$
**procedure** SEARCH$((s,h), b(\theta))$
    **for** $e = 1 \cdots E$ **do**
        SIMULATE$((s,h), b(\theta), d_{\max})$
    **end for**
    $v_{\text{ret}} = \sum_{a \in C((s,h))} \frac{N((s,h),a)}{N((s,h))} Q((s,h), a)$
    **return** $\pi_{\text{ret}}, v_{\text{ret}}$
**end procedure**
**procedure** SIMULATE$((s,h), b(\theta), d)$
    **if** $d == 0$ **then return** $V((s,h))$
    $a \leftarrow$ ACTIONPW$((s,h))$
    $r, s', b'(\theta) \leftarrow$ STATEPW$((s,h), b(\theta), a)$
    $N((s,h)) \mathrel{+}= 1, N((s,h),a) \mathrel{+}= 1$
    **if** $N((s,h)) > 1$ **then**
        $R \leftarrow$ SIMULATE$((s', hars'), b'(\theta), d-1)$
    **else**
        $R \leftarrow V((s', hars'))$
    **end if**
    Access $\tilde{r}$ or calculate $\tilde{r}$ using Eq. (5)
    $R \leftarrow \tilde{r} + \gamma R$, cache $\tilde{r}$
    $Q((s,h), a) \mathrel{+}= \frac{R - Q((s,h),a)}{N((s,h),a)}$
    **return** $R$
**end procedure**

**procedure** ACTIONPW$((s,h))$
    **if** first visit **then** $C((s,h)) \leftarrow \emptyset$
    **if** $\lfloor N((s,h))^\alpha \rfloor \geq |C((s,h))|$ **then**
        $a \sim \pi(\cdot | (s,h))$
        $C((s,h)) \leftarrow C((s,h)) \cup \{a\}$
        $N((s,h),a), Q((s,h),a) \leftarrow 0, 0$
    **else**
        $a \leftarrow \arg\max_{x \in C((s,h))} \tilde{Q}((s,h), x)$
    **end if**
    **return** $a$
**end procedure**
**procedure** STATEPW$((s,h), b(\theta), a)$
    **if** first visit **then** $C((s,h), a) \leftarrow \emptyset$
    **if** $\lfloor N((s,h),a)^\beta \rfloor \geq |C((s,h),a)|$ **then**
        $r \sim \sum_{i=1}^K b(\theta)(i) \mathcal{R}_\theta^i(\cdot | s, a)$
        $s' \sim \sum_{i=1}^K b(\theta)(i) \mathcal{P}_\theta^i(\cdot | s, a)$
        Update $b(\theta)$ to $b'(\theta)$ using Eq. (4)
        $C((s,h), a) \leftarrow C((s,h), a) \cup \{(r, s', b'(\theta))\}$
        $N((s', hars')) \leftarrow 0$
        **return** $r, s', b'(\theta)$
    **end if**
    **return** the least visited node in $C((s,h), a)$
**end procedure**

---

the learned policy may overfit to an inaccurate model (by overestimating the expected return) and fail in the true MDP. Although the agent could adapt its belief and follow more reliable ensemble members in the Bayesian RL framework, there could be regions in the state-action space where none of the members generalize well, as they are all learned from a static offline dataset. A typical solution is to construct a P-MDP (see Section 3), which lower-bounds the true MDP and discourages the policy from regions where there is large discrepancy between the true and learned world models. We construct the P-MDP by modifying each reward estimation $r$ into $\tilde{r}$: $(\mu_\theta(s, a) = \sum_{i=1}^K b(\theta)(i) \mu_\theta^i(s, a))$

$$\tilde{r}(s, a, r, b(\theta)) = r - \lambda \sqrt{\sum_{i=1}^K b(\theta)(i)(\sigma_\theta^i(s, a)^2 + \mu_\theta^i(s, a)^2) - \mu_\theta(s, a)^2} \tag{5}$$

The reward penalty is weighted by a hyperparameter $\lambda > 0$ and corresponds to the standard deviation (std) of the mixture of Gaussian dynamics models, where $\mu_\theta^i$ and $\sigma_\theta^i$ are the mean and std from the ensemble member $i$. This penalty design combines epistemic and aleatoric model uncertainty and has been shown to be successful at capturing errors in predicted dynamics (Lu et al. (2022))[2].

## 4.2 BAYES ADAPTIVE MCTS IN CONTINUOUS STATE AND ACTION SPACES

BAMCP (Guez et al. (2013)) has been successful in solving large-scale BAMDPs, as detailed in Appendix A, but it is limited to scenarios with discrete state and action spaces. In this subsection, we introduce a novel planning method to approximate the Bayes-optimal policy at a decision point $(s, h)$ ($h$ denotes the transition history that ends at $s$), which can be used to solve BAMDPs with continuous states/actions and stochastic transition kernels.

**Double Progressive Widening (DPW):** DPW (Couëtoux et al. (2011); Auger et al. (2013)) is a technique to extend the use of MCTS to continuous state and action spaces. Instead of exploring all possible actions and next states, DPW maintains a finite list of options to search at each decision

---

[2]In the original literature (Lakshminarayanan et al. (2017); Lu et al. (2022)), the ensemble is treated as a uniformly-weighted mixture model, i.e., $b(\theta)(i) = 1/K$, $(i = 1, \cdots K)$, since their belief is not adaptive. Equation (5) fits into the Bayesian RL framework by adapting the belief $b(\theta)$, which is part of our novelty.

point, incrementally adding new options to the list based on the visitation counts of that decision point. To be specific, for a node $(s, h)$, a new action $a$ is sampled (with the current policy $\pi$) and added to its children set $C((s, h))$, if $\lfloor N((s, h))^\alpha \rfloor \geq |C((s, h))|$, where $\alpha \in (0, 1)$ is a hyperparameter that controls the growth rate and $N$ denotes the visitation counts of $(s, h)$. Otherwise, an action is selected from the existing children set $C((s, h))$ according to the UCT (Kocsis & Szepesvári (2006)) rule. Similarly, to handle the infinitely many possible state transitions, a new next state $s'$ is added to the children set $C((s, h), a)$ only if $\lfloor N((s, h), a)^\beta \rfloor \geq |C((s, h), a)|$ ($\beta \in (0, 1)$). Otherwise, the least visited child in $C((s, h), a)$ will be selected as the next state. With DPW, the sets of possible actions or next states to explore are finite, allowing deep tree search as in discrete scenarios. The more promising states and actions (with higher $N$) have more subsequent branches, thereby reducing corresponding estimation uncertainty.

**Integration of DPW and BAMCP:** Directly combining DPW and BAMCP (i.e., Algorithm 3) cannot solve BAMDPs with continuous state and action spaces. As introduced in Appendix A, BAMCP relies on root sampling, which samples dynamics functions only at the root node and follows a specific dynamics function throughout a simulation rollout. However, the rationale of root sampling (i.e., Lemma A.1) does not hold when applying DPW[3]. As an alternative design, Polynomial Upper Confidence Tree (PUCT, Auger et al. (2013)), built upon DPW, is a provably consistent planning method for solving MDPs with infinite-scale state and action spaces and highly stochastic transition dynamics. Thus, we propose casting BAMDPs into MDPs (i.e., $\mathcal{M}^+$ defined in Section 2) and solving them with PUCT. The pseudo code is shown as Algorithm 1. Ideally, as the number of simulations $E \to \infty$, PUCT can find a near-optimal solution of $\mathcal{M}^+$ (Auger et al. (2013)), which is also a near Bayes-optimal solution for the true environment.

**Proposed algorithm – Continuous BAMCP:** In Algorithm 1, each simulation follows a path from the root node to an unvisited node, utilizing progressive widening when sampling actions or next states, as detailed in the ACTIONPW and STATEPW procedures. Compared to PUCT, the significant modifications include: (1) replacing $\langle \mathcal{S}, \mathcal{P}, \mathcal{R} \rangle$ in MDPs with their extended definitions in BAMDPs, i.e., $\langle \mathcal{S}^+, \mathcal{P}^+, \mathcal{R}^+ \rangle$, and (2) applying reward penalties to account for model uncertainty. To be specific, in STATEPW, $r$ and $s'$ are sampled from the distribution predicted by all ensemble members, which is a practical implementation of sampling from $\mathcal{R}^+$ and $\mathcal{P}^+$ as outlined in Eq. (1). After receiving the transition $(s, a, r, s')$, the belief vector $b(\theta)$ is updated to $b'(\theta)$ following Eq. (4), finishing the transition in $\mathcal{S}^+$ from $(s, b(\theta))$ to $(s', b'(\theta))$. Meanwhile, the transition history $h$ is updated to $h' = hars'$. Secondly, in SIMULATE, the reward $r$ is adjusted with a penalty term defined in Eq. (5), which is then used to calculate the return $R$. Applying such a reward penalty can effectively mitigate the issue of model exploitation.

Algorithm 1 can be used to approximate the Bayes-optimal policy at $(s, h)$, which is $\pi_{\text{ret}}(a|(s, h)) \propto N((s, h), a)), a \in C((s, h))$ (Auger et al. (2013)). However, we aim to solve the entire BAMDP offline, eliminating the need for anything beyond simple inference using the policy network during deployment. This necessitates a well-learned policy function at each decision point, but we cannot execute Algorithm 1 at every $(s, h)$ due to the scale of the state space. Therefore, we integrate the planning algorithm into a policy iteration framework as introduced in the next subsection. In this case, $\pi$ and $V$ in Algorithm 1 denote the policy and value functions from the previous learning iteration[4]; while $\pi_{\text{ret}}$ and $v_{\text{ret}}$ are the improved policy and value estimates for specific decision points. As additional details, multiple terms (labeled in blue) in Algorithm 1 have alternative designs across different literatures, which we elaborate on in Appendix B.

### 4.3 THE OVERALL FRAMEWORK: SEARCH-BASED POLICY ITERATION

In this subsection, we present how to integrate continuous BAMCP into policy improvement and policy evaluation. By iteratively running these procedures, we can approach a near Bayes-optimal policy, i.e., $\pi$, that can be directly referred to during execution in the true environment. The pseudo code of the overall framework is shown as Algorithm 2. For efficiency, a learner and a number of

---

[3]The last equality of Eq. (6) does not hold, since $\tilde{b}(\theta|has') \propto \tilde{b}(\theta|ha)\widetilde{\mathcal{P}}_\theta(s'|s, a) \neq \tilde{b}(\theta|ha)\mathcal{P}_\theta(s'|s, a)$. $\widetilde{\mathcal{P}}_\theta(s'|s, a)$ represents the state transition distribution when applying DPW, which differs from the true distribution $\mathcal{P}_\theta(s'|s, a)$, as dictated by the DPW rule.

[4]$\pi$ and $V$ are functions of $(s, h)$ because the states in BAMDPs consist of both $s$ and the corresponding belief $b(\theta)$, with $b(\theta)$ being a function of the history $h$ (according to its recursive updating rule).

---

**Algorithm 2** Search-based Policy Iteration

**Input:** $T, E_l, \mathcal{P}_\theta^{1:K}, \mathcal{R}_\theta^{1:K}$

Initialize $\pi$ and $V$, $\mathcal{D} \leftarrow \emptyset$
**procedure** LEARNER
    $e \leftarrow 0$
    **while** true **do**
        $\{(s, h, \pi_{\text{ret}}, \tilde{r}, s', h')_i\}_{i=1}^{B} \sim \mathcal{D}$
        $\min_\pi \mathcal{L}(\pi, \{(s, h, \pi_{\text{ret}})_i\}_{i=1}^{B})$
        $\min_V \mathcal{L}^{\text{TD}}(V, \{(s, h, \tilde{r}, s', h')_i\}_{i=1}^{B})$
        $e += 1$
        Update $\pi, V$ in ACTOR if $e\%E_l == 0$
    **end while**
**end procedure**
$\mathcal{L}(\pi, \{\tau_i\}_{i=1}^{B}) = -\sum_{((s,h), \pi_{\text{ret}})} \pi_{\text{ret}}^T \log \pi(\cdot|(s, h))/(BT)$

**procedure** ACTOR
    **while** true **do**
        Sample $s$ from $\mathcal{D}_\mu, h \leftarrow s, \tau \leftarrow []$
        Obtain the prior $b(\theta)$ at $s$
        **for** $t = 1 \cdots H$ **do**
            $\pi_{\text{ret}}, v_{\text{ret}} \leftarrow \text{SEARCH}((s, h), b(\theta))$
            $a \sim \pi_{\text{ret}}(\cdot|(s, h))$
            Acquire $r, s', b'(\theta)$ as in STATEPW
            Calculate $\tilde{r}$ using Eq. (5)
            Append $\tau$ with $((s, h), \tilde{r}, \pi_{\text{ret}}, v_{\text{ret}})$
            $s, h, b(\theta) \leftarrow s', hars', b'(\theta)$
        **end for**
        $\mathcal{D} \leftarrow \mathcal{D} \cup \{\tau\}$
    **end while**
**end procedure**

---

actors execute in parallel, reading from and sending data to the replay buffer $\mathcal{D}$ respectively. The actors update their copies of policy and value functions every $E_l$ learner steps.

Each actor interacts with the learned world models to sample trajectory segments $\tau$. The starting states of these segments are sampled from the provided dataset $\mathcal{D}_\mu$. Notably: (1) the segment length $H$ is kept relatively short to minimize error accumulation when interacting with the learned world models; and (2) the prior belief at a starting state $s$ is obtained by performing the Bayesian posterior update (i.e., Eq. (4)) on the offline trajectory to which $s$ belongs. These belief updates are reliable, as the offline trajectories are collected from the real environment. At each time step of the segment, a SEARCH procedure (defined in Algorithm 1) is executed at the current decision point $(s, h)$. The search result $\pi_{\text{ret}}$ is then used to indicate the action choice, i.e., $a \sim \pi_{\text{ret}}(\cdot|(s, h))$. As in STATEPW, the subsequent transition process follows a BAMDP, where $r \sim \mathcal{R}^+(\cdot|(s, h), a)$, $s' \sim \mathcal{P}^+(\cdot|(s, h), a)$, and the belief $b(\theta)$ is adapted with the new transition. The collected segments are used in the learning process, where $\pi$ is trained to mimic the search result $\pi_{\text{ret}}$ by minimizing a cross-entropy loss (i.e., $\mathcal{L}(\pi, \{\tau_i\}_{i=1}^{B})$), while $V$ is updated using standard temporal difference learning methods (e.g., SAC (Haarnoja et al. (2018))) based on the sampled transitions.[5] As noted in (Hubert et al. (2021)), $\pi_{\text{ret}}$ improves $\pi$ at each decision point, so repeatedly applying continuous BAMCP to obtain $\pi_{\text{ret}}$ and projecting the search results to the parameter space of $\pi$ (through supervised learning) constitute a powerful improvement operator to iteratively enhance the policy $\pi$.

## 5 EVALUATION

Our experiments target at the following research questions: (**RQ1**) Would using BAMDP improve policy performance (by properly adapting the belief over the ensemble members)? (**RQ2**) Can the proposed search method, Continuous BAMCP, further enhance the performance of MBRL? (**RQ3**) How can the search outcomes (i.e., $\pi_{\text{ret}}$) be effectively used for policy updates? (**RQ4**) Is it necessary to apply reward penalties to mitigate the overestimation issue? (**RQ5**) Does the proposed algorithm outperform other deep-search-based offline RL methods, such as MuZero and its variants? (**RQ6**) Can the proposed algorithm be applied to complex, real-world data-driven control tasks? We address the first four research questions in Section 5.1, **RQ5** in Section 5.2, and **RQ6** in Section 5.3.

### 5.1 BENCHMARKING RESULTS ON D4RL MUJOCO TASKS

To evaluate the effectiveness of each component in our algorithm design, we introduce three variants: (1) **BA-MBRL** leverages learned world models as surrogate simulators, applying reward penalties to collected transitions and using standard online RL algorithms (e.g., SAC) to learn a policy. While

---

[5] The search result $v_{\text{ret}}$ can be used to construct the value target; however, we did not observe empirical performance improvements from it.

| Data Type | Environment | BA-MCTS-SL (ours) | BA-MCTS (ours) | BA-MBRL (ours) | Optimized | COMBO | MOReL | MOPO |
|---|---|---|---|---|---|---|---|---|
| random | HalfCheetah | $29.20 \pm 2.00$ | $36.23 \pm 1.04$ | $32.76 \pm 1.16$ | 31.7 | **38.8** | 25.6 | 35.4 |
| random | Hopper | $33.83 \pm 0.10$ | $31.56 \pm 0.12$ | $31.47 \pm 0.03$ | 12.1 | 17.9 | **53.6** | 11.7 |
| random | Walker2d | $21.89 \pm 0.07$ | $21.59 \pm 0.32$ | $21.45 \pm 0.53$ | 21.7 | 7.0 | **37.3** | 13.6 |
| medium | HalfCheetah | $70.47 \pm 3.52$ | **75.84** $\pm 3.81$ | $56.54 \pm 5.20$ | 45.7 | 54.2 | 42.1 | 42.3 |
| medium | Hopper | $97.75 \pm 7.09$ | $96.70 \pm 14.0$ | **98.25** $\pm 3.42$ | 69.3 | 97.2 | 95.4 | 28.0 |
| medium | Walker2d | **82.24** $\pm 1.85$ | $74.73 \pm 3.25$ | $75.41 \pm 4.17$ | 79.7 | 81.9 | 77.8 | 17.8 |
| med-replay | HalfCheetah | $61.16 \pm 1.60$ | **65.45** $\pm 0.81$ | $62.50 \pm 0.18$ | 58.0 | 55.1 | 40.2 | 53.1 |
| med-replay | Hopper | **106.3** $\pm 0.13$ | $101.8 \pm 3.46$ | $93.91 \pm 4.25$ | 90.8 | 89.5 | 93.6 | 67.5 |
| med-replay | Walker2d | $92.13 \pm 5.13$ | $95.06 \pm 2.11$ | **97.54** $\pm 1.93$ | 65.8 | 56.0 | 49.8 | 39.0 |
| med-expert | HalfCheetah | $80.53 \pm 6.63$ | $76.16 \pm 10.3$ | $90.52 \pm 4.13$ | **104.2** | 90.0 | 53.3 | 63.3 |
| med-expert | Hopper | **112.2** $\pm 0.29$ | $108.3 \pm 0.22$ | $107.8 \pm 0.37$ | 105.8 | 111.1 | 108.7 | 23.7 |
| med-expert | Walker2d | $107.7 \pm 0.82$ | **110.0** $\pm 1.74$ | $84.71 \pm 0.87$ | 97.1 | 103.3 | 95.6 | 44.6 |
| Average Score | | **74.62** | 74.45 | 71.06 | 65.16 | 66.83 | 64.42 | 36.67 |

Table 1: Comparisons between the proposed algorithms and SOTA offline model-based RL methods on the D4RL benchmark suite. Each value represents the normalized score, as proposed in (Fu et al. (2020)), of the policy trained by the corresponding algorithm. These scores are undiscounted returns normalized to approximately range between 0 and 100, where a score of 0 corresponds to a random policy and a score of 100 corresponds to an expert-level policy. For our algorithms, we report the average score of the final ten policy learning epochs and its standard deviation across three random seeds. Results in the last four columns are taken from the original papers (Lu et al. (2022); Yu et al. (2021); Kidambi et al. (2020); Yu et al. (2020)), respectively.

following existing offline MBRL methods, it models the problem as a BAMDP (rather than an MDP), with environment transitions defined by Equations (1) and (4) and the reward penalty by Equation (5), and is designed to evaluate the effectiveness of Bayesian RL. (2) **BA-MCTS** builds on BA-MBRL by introducing Continuous BAMCP (Algorithm 1) to plan at decision points, rather than inferring directly from the policy, to generate trajectories for downstream SAC, demonstrating the impact of deep search on policy learning. (3) **BA-MCTS-SL**, described in Algorithm 2, replaces the policy learning algorithm in BA-MCTS from policy gradient methods (as in SAC) with supervised learning (SL), allowing us to compare which approach offers a more efficient policy update mechanism, particularly for continuous control tasks.

We first evaluate our algorithms on a widely-used continuous control benchmark for offline RL methods – D4RL MuJoCo (Fu et al. (2020)). The evaluation results for three types of robotic agents, each with offline datasets of four different qualities, are presented in Table 1. (1) Compared to SOTA offline MBRL methods, our algorithms achieve superior performance on nine out of twelve tasks. In terms of average performance, BA-MBRL significantly outperforms the baselines, demonstrating the effectiveness of using BAMDPs to handle model uncertainties in offline MBRL and addressing **RQ1**. Further, in Appendix H, we show that employing a Bayes-adaptive ensemble, instead of a uniform ensemble, improves the prediction likelihood for the provided offline trajectories and reduces the prediction errors in imaginary rollouts. As illustrative examples of such performance improvement, Figure 4 tracks belief adaptation during an offline rollout and several imaginary rollouts. (2) Both BA-MCTS and BA-MCTS-SL further improve upon BA-MBRL, highlighting the enhancement brought by deep search in policy learning, as related to **RQ2**. Notably, we apply Continuous BAMCP to only 10% of states when collecting training trajectories, while for the remaining states, we sample actions directly from the policy, i.e., $a \sim \pi(\cdot|s)$. Increasing the search ratio could further enhance policy performance at the cost of increased computation. (3) For **RQ3**, BA-MCTS-SL performs similarly to BA-MCTS, validating the effectiveness of both policy update mechanisms. However, BA-MCTS-SL struggles on Walker2d, where a warm-up training phase (using BA-MBRL) is required to establish a better initial policy. On the other hand, the advantage of the SL-based policy update is evident in the training plots of our algorithms in Figure 2, where BA-MCTS-SL exhibits much smoother learning curves compared to the other two algorithms, indicating greater robustness in model selection. (4) We further compare our algorithms with model-free offline policy learning methods, as shown in Appendix D. The performance improvement is even greater than that over model-based methods, highlighting the necessity of model-based learning. Particularly, when data quality is low, merely mimicking or staying close to the behavior policy would result in an underperforming policy. (5) To investigate **RQ4**, we provide an ablation study in Appendix F to demonstrate the necessity of incorporating the reward penalty in offline MBRL to prevent the overexploitation of inaccurate world models. Additionally, we find that the SL-based policy update is less sensitive to model inaccuracies.

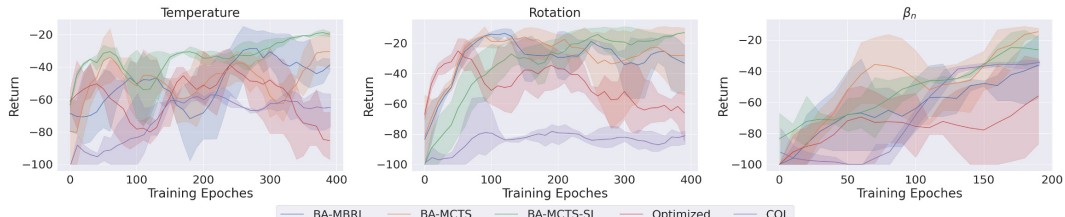

Figure 1: Evaluation results for the tokamak control tasks. The figure shows the change in episodic returns over training epochs for the proposed algorithms and baselines across three target tracking tasks in the nuclear fusion scenario. Solid lines represent the average performance, while shaded areas indicate the 95% confidence intervals.

For fair comparisons and real-time execution, we do not perform test-time search and adopt the same policy architecture as the baselines, i.e., a feedforward neural network, rather than an RNN that incorporates transition history as input. These alternative designs have the potential to further improve our algorithms. For implementation, we build on the codebase of Optimized (Lu et al. (2022)), which thoroughly explores design choices in offline MBRL, making minimal changes to the code and hyperparameter settings[6]. Therefore, we believe the performance improvements stem from the Bayesian RL framework and deep search components. Both components can be integrated with other advancements in offline MBRL, such as more accurate world model learning and improved uncertainty quantification for constructing P-MDPs. To validate this, we incorporate a reward penalty design proposed in (Sun et al. (2023)) with BA-MCTS, achieving improved performance in several benchmarking tasks and demonstrating its SOTA performance (see Appendix I).

### 5.2 Comparison with MuZero-type Methods

MuZero also applies deep search to MBRL. To evaluate its performance on D4RL MuJoCo tasks and answer **RQ5**, we use the open-source implementation and hyperparameter configurations of Sampled EfficientZero (Ye et al. (2021)) provided by LightZero (Niu et al. (2023)). Benchmarking results from LightZero indicate that Sampled EfficientZero achieves the best performance on (online) MuJoCo locomotion tasks compared to other MuZero variants. To adapt Sampled EfficientZero for offline learning, we employ the reanalyse technique proposed by (Schrittwieser et al. (2021)). The evaluation results are presented in Figure 3 (Appendix E). For reference, the expert-level episodic returns (corresponding to scores of 100) for HalfCheetah, Hopper, and Walker2d are 12135, 3234.3, and 4592.3, respectively. As shown, the results are significantly worse compared to the performance of offline RL methods listed in Table 1, despite Sampled EfficientZero's higher computational cost. (In Appendix E, we provide a detailed comparison of the computational costs of our proposed algorithms and Sampled EfficientZero.) Notably, both Sampled EfficientZero and BA-MCTS-SL rely on supervised learning for policy improvement. However, for continuous control tasks, the agent can only sample a finite number of actions at a decision point, and the search result (e.g., $\pi_{\text{ret}}$ in Algorithm 1) is a distribution over this finite set, which could be a poor approximation of the optimal action distribution. Thus, purely mimicking the search result may be less sample-efficient than policy gradient methods, as it fails to account for the continuous nature of the action space. Further, world model learning is the foundation of MBRL and can be particularly challenging in **continuous control and offline learning settings**, where the state-action space is vast but training data is limited. Sampled EfficientZero integrates model learning and policy training into a single stage, which significantly increases the learning difficulty (compared to BA-MCTS-SL).

### 5.3 Applications to Tokamak Control

Finally, to investigate **RQ6**, we evaluate our algorithms on three target tracking tasks in tokamak control. The tokamak is one of the most promising confinement devices for achieving controllable nuclear fusion, where the primary challenge lies in confining the plasma, i.e., an ionized gas of hydrogen isotopes, while heating it and increasing its pressure to initiate and sustain fusion reactions (Pironti & Walker (2005)). Tokamak control involves applying a series of direct actuators (e.g.,

---

[6]The detailed hyperparameter setups of our algorithms are provided in Appendix C.

| Task | BA-MCTS -SL (ours) | BA-MCTS (ours) | BA-MBRL (ours) | CQL | Optimized |
|---|---|---|---|---|---|
| Temperature | **-21.16** $\pm$ 5.00 | -23.83 $\pm$ 9.66 | -29.35 $\pm$ 4.72 | -59.62 $\pm$ 1.57 | -83.55 $\pm$ 10.56 |
| Rotation | **-14.14** $\pm$ 1.88 | -19.07 $\pm$ 5.85 | -31.33 $\pm$ 11.54 | -85.48 $\pm$ 2.72 | -71.54 $\pm$ 9.88 |
| $\beta_n$ | -37.03 $\pm$ 17.98 | **-18.93** $\pm$ 1.75 | -23.4 $\pm$ 10.77 | -36.37 $\pm$ 1.17 | -57.84 $\pm$ 10.27 |
| Average | -24.11 | **-20.61** | -28.03 | -60.49 | -70.98 |

Table 2: Comparisons between the proposed algorithms and offline RL baselines on the target tracking tasks. For each algorithm, we report the average return of the final ten policy learning epochs and its standard deviation across three different random seeds.

neutral beam, ECH power, magnetic field) and indirect actuators (e.g., setting targets for the plasma shape and density) to confine the plasma to achieve a desired state or track a given target. This sophisticated physical process is an ideal test bed for our algorithms. Specifically, we use a well-trained data-driven dynamics model provided by Char et al. (2024) as a "ground truth" simulator for the nuclear fusion process during evaluation, and generate a dataset containing 725270 transitions using this model for offline RL. We select a reference shot (i.e., an episode of a fusion process) from DIII-D[7], and use its trajectories of Ion Rotation, Electron Temperature, and $\beta_n$ as targets for three tracking tasks. These are critical quantities in tokamak control, particularly $\beta_n$, which serves as an economic indicator of the efficiency of nuclear fusion. The tracking tasks have a 28-dimensional state space and a 14-dimensional action space, both continuous. Moreover, these tasks are **highly stochastic**, as the underlying dynamics model is a probabilistic neural network and each state transition is a sample from this model. For details on the simulator, and the design of the state/action spaces and reward functions, please refer to Appendix G. We compare our algorithms with SOTA model-free and model-based offline RL methods, specifically CQL and Optimized. The learning performance on the three tracking tasks is shown in Figure 1, where the x-axis and y-axis represent the training epochs and (negative) full-shot tracking errors, respectively. Our algorithms consistently outperform the baselines. Notably, the offline dataset does not include the reference shot or any similar, nearby shots. Therefore, restricting the policy to stay close to the behavior policy, as done in model-free methods, can be problematic. Also, learning dynamics models for MBRL is quite challenging in this nuclear fusion scenario. Our algorithms share the same ensemble of dynamics models with "Optimized" for policy learning, and the comparisons can demonstrate the superiority of Bayesian RL and deep search. Figure 1 has been smoothed for visualization[8]. We further report the average return over the final 10 training epochs in Table 2, and the conclusions align with those from the D4RL MuJoCo tasks, showing the robustness of our proposed algorithms.

## 6 CONCLUSION AND DISCUSSIONS

In this work, we propose framing offline model-based reinforcement learning (MBRL) as a Bayes Adaptive Markov Decision Process (BAMDP) to better address uncertainties in the world models learned from offline datasets. We also introduce a novel planning method for solving BAMDPs in continuous state and action spaces using Monte Carlo Tree Search. This planning process is integrated into a policy iteration framework, enabling the derivation of a policy suitable for real-time execution from the planning results. In our evaluation, we test several variants of our algorithms to separately highlight the effectiveness of Bayesian RL and deep search. Additionally, we compare two different approaches for policy updates (based on the search results) in continuous control tasks: supervised learning and policy gradient methods. Our findings demonstrate that: (1) adapting beliefs over an ensemble of world models based on experience yields more accurate model approximations for MBRL; (2) deep search improves learning performance by incorporating planning and additional computation input; and (3) while supervised-learning-based policy updates result in smoother learning curves, they may struggle in complex continuous control tasks due to their approximation of the continuous action space as a finite set of action samples. For future work, our algorithms can be improved by integrating advancements in offline MBRL and Bayesian RL, such as Bayesian Neural Networks, techniques to address sparse rewards in MBRL, and more principled approaches to construct pessimistic MDPs beyond those based on ensemble discrepancy.

---

[7]DIII-D is a tokamak device located in San Diego, California, operated by General Atomics.

[8]The episodic return is plotted every 10 training epochs, with the y-axis representing the average value of a sliding window of length 5.

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

# A   BAMCP

---

**Algorithm 3** BAMCP

---

**Input:** $\pi, E, d_{\max}, \mathcal{R}, \gamma, \mathcal{A}, c$

**procedure** SEARCH$((s, h), b(\theta))$
    **for** $e = 1 \cdots E$ **do**
        $\theta \sim b(\cdot)$
        SIMULATE$((s, h), \theta, d_{\max})$
    **end for**
    **return** $\arg\max_a Q((s, h), a)$
**end procedure**

**procedure** ROLLOUT$((s, h), \theta, d)$
    **if** $d == 0$ **then**
        **return** $0$
    **end if**
    $a \sim \pi(\cdot|(s, h))$
    $s' \sim \mathcal{P}_\theta(\cdot|s, a)$
    $r \leftarrow \mathcal{R}(s, a)$
    $R \leftarrow r + \gamma \text{ROLLOUT}((s', has'), \theta, d - 1)$
    **return** $R$
**end procedure**

**procedure** SIMULATE$((s, h), \theta, d)$
    **if** $d == 0$ **then**
        **return** $0$
    **end if**
    **if** $N((s, h)) == 0$ **then**
        **for** $a \in \mathcal{A}$ **do**
            $N((s, h), a), Q((s, h), a) \leftarrow 0, 0$
        **end for**
        $a \sim \pi(\cdot|(s, h))$
        $s' \sim \mathcal{P}_\theta(\cdot|s, a), r \leftarrow \mathcal{R}(s, a)$
        $R \leftarrow r + \gamma \text{ROLLOUT}((s', has'), \theta, d - 1)$
        $N((s, h)), N((s, h), a) \leftarrow 1, 1$
        $Q((s, h), a) \leftarrow R$
        **return** $R$
    **end if**
    $a \leftarrow \arg\max_x Q((s, h), x) + c\sqrt{\frac{\log N((s, h))}{N((s, h), x)}}$
    $s' \sim \mathcal{P}_\theta(\cdot|s, a), r \leftarrow \mathcal{R}(s, a)$
    $R \leftarrow r + \gamma \text{SIMULATE}((s', has'), \theta, d - 1)$
    $N((s, h)) \mathrel{+}= 1, N((s, h), a) \mathrel{+}= 1$
    $Q((s, h), a) \mathrel{+}= \frac{R - Q((s, h), a)}{N((s, h), a)}$
    **return** $R$
**end procedure**

---

Bayes Adaptive Monte Carlo Planning (BAMCP, Guez et al. (2013)) is a sample-based online planning method, aiming to find the action $a^*$ that approximately maximizes the expected return at a decision point $(s, h)$ under the BAMDP. Its detailed pseudo code is shown as Algorithm 3. BAMCP has demonstrated success in solving BAMDPs with large-scale **discrete** state and action spaces. Its key algorithmic ideas include: (1) applying MCTS with an efficient exploration strategy – UCT (Kocsis & Szepesvári (2006)) to the BAMDP in order to simulate the outcomes of different action choices; (2) utilizing root sampling to avoid frequent Bayesian posterior updates. Specifically, the UCT rule is used for selecting actions at non-leaf nodes, i.e., $a \leftarrow \arg\max_x Q((s, h), x) + c\sqrt{\frac{\log N((s, h))}{N((s, h), x)}}$, managing the tradeoff between exploration and exploitation. Root sampling refers to sampling the dynamics model only at the root node (i.e., $\theta \sim b(\cdot)$) and not adapting the belief $b(\cdot)$ according to the Bayes rule during the search process, of which the rationality is justified in the following lemma.

**Lemma A.1.** *For all suffix histories $h'$ of $h$, $b(\theta|h') = \tilde{b}(\theta|h')$. Here, $b(\theta|h')$ is the true posterior probability of $\theta$ at the decision point $h'$, while $\tilde{b}(\theta|h')$ is the probability of experiencing $\theta$ at $h'$ when using root sampling.*

*Proof.* This lemma can be proved by induction.

Base case: When $h' = h$, $b(\theta|h') = \tilde{b}(\theta|h') = b(\theta)$.

Step case:

$$
\begin{aligned}
b(\theta|has') &= P(has'|\theta)P(\theta)/P(has') \\
&= P(h|\theta)\mathcal{P}_\theta(s'|s, a)P(\theta)/P(has') \\
&= b(\theta|h)P(h)\mathcal{P}_\theta(s'|s, a)/P(has') \\
&= Zb(\theta|h)\mathcal{P}_\theta(s'|s, a) \\
&= Z\tilde{b}(\theta|h)\mathcal{P}_\theta(s'|s, a) \\
&= Z\tilde{b}(\theta|ha)\mathcal{P}_\theta(s'|s, a) = \tilde{b}(\theta|has')
\end{aligned}
\tag{6}
$$

Here, $Z = 1/\int_\theta \mathcal{P}_\theta(s'|s,a)b(\theta|h)d\theta = 1/\int_\theta \mathcal{P}_\theta(s'|s,a)\tilde{b}(\theta|h)d\theta = 1/\int_\theta \mathcal{P}_\theta(s'|s,a)\tilde{b}(\theta|ha)d\theta$ is the normalization constant. The fifth equality in Eq. (6) holds due to the inductive hypothesis. The sixth equality is based on the fact that the choice of $a$ at each node $h$ is made independently of the sample $\theta$. As for the last equality, to experience $\theta$ at $has'$, the sample $\theta$ needs to traverse $ha$ (with probability $\tilde{b}(\theta|ha)$) and then the state $s'$ needs to be sampled, which is with probability $\mathcal{P}_\theta(s'|s,a)$, so $\tilde{b}(\theta|has') \propto \tilde{b}(\theta|ha)\mathcal{P}_\theta(s'|s,a)$. □

## B  ALTERNATIVE DESIGN CHOICES FOR CONTINUOUS BAMCP

The terms labeled in blue in Algorithm 1 have alternative design choices. Empirical comparisons among these alternatives are reserved for future work.

$\tilde{Q}((s,h),x)$ in the exploration strategy, i.e., $a \leftarrow \arg\max_{x \in C((s,h))} \tilde{Q}((s,h),x)$, could take various forms. For instance, in (Couëtoux et al. (2011); Guez et al. (2013); Lee et al. (2020)), $\tilde{Q}((s,h),x) = Q((s,h),x) + c\sqrt{\frac{\log N((s,h))}{N((s,h),x)}}$; in PUCT (Auger et al. (2013)), $\tilde{Q}((s,h),x) = Q((s,h),x) + \sqrt{\frac{N((s,h))^{e(d)}}{N((s,h),x)}}$, where $e(d)$ is a schedule of coefficients related to the search depth $d$; in Sampled MuZero (a variant of MuZero that can be applied in continuous action spaces (Hubert et al. (2021))), $\tilde{Q}((s,h),x) = Q((s,h),x) + \hat{\pi}(x|(s,h))\frac{\sqrt{N((s,h))}}{1+N((s,h),x)}\left(c_1 + \log\left(\frac{N((s,h))+c_2+1}{c_2}\right)\right)$. Here, $c, c_1, c_2$ are hyperparameters, $\hat{\pi} = \hat{\beta}\pi^{1-1/\tau}$ is a sample policy defined upon the real policy $\pi$. In particular, at each decision point $(s,h)$, Sampled MuZero would sample $M$ actions $\{a_i\}$ from the distribution $\pi^{1/\tau}$ and accordingly define $\hat{\beta}(a|(s,h)) = \sum_i \mathbb{1}_{a_i=a}/M$, where $\tau > 0$ is a temperature hyperparameter. Thus, Sampled MuZero does not adopt progressive widening like ours. Following BAMCP, we adopt the first definition of $\tilde{Q}((s,h),x)$, though it could potentially be improved with other choices. In addition, as an implementation trick (Hamrick et al. (2021)), the Q estimates are usually normalized into $\bar{Q} \in [0,1]$ before being used to calculate $\tilde{Q}$ as above. The normalized estimates can be computed as $\bar{Q}((s,h),x) = \frac{Q((s,h),x)-Q_{\min}}{Q_{\max}-Q_{\min}}$, where $Q_{\max}$ and $Q_{\min}$ are the maximum and minimum Q values observed in the search tree so far.

As the planning/search result, $\pi_{\text{ret}}$ can take multiple forms. In (Guez et al. (2013); Sunberg & Kochenderfer (2018); Lee et al. (2020)), $\pi_{\text{ret}}((s,h)) = \arg\max_{a \in C((s,h))} Q((s,h),a)$; in (Sampled) MuZero, $\pi_{\text{ret}}(a|(s,h)) = \frac{N((s,h),a)^{1/\tau}}{\sum_{x \in C((s,h))} N((s,h),x)^{1/\tau}}$; in ROSMO (a variant of MuZero with improved performance in offline scenarios (Liu et al. (2023))), $\pi_{\text{ret}}(a|(s,h)) \propto \pi(a|(s,h))\exp(Q((s,h),a) - V((s,h)))$. Here, $\tau \in (0,1]$ is a temperature parameter and decays with the training process, ensuring the action selection becomes greedier. We select the second form for $\pi_{\text{ret}}$ in Algorithm 1. This is because (1) as described in PUCT, the returned action should be the most visited one, which is not necessarily the one with the highest Q value, and (2) ROSMO adopts one-step look-ahead rather than deep tree search at each root node, which does not align with our approach.

As for the conditions of double progressive widening, PUCT designs $\alpha$ and $\beta$ to be functions of the search depth $d$, while UCT-DPW (Couëtoux et al. (2011)) utilizes a different set of conditions: $\lceil K_a N((s,h))^\alpha \rceil \geq |C((s,h))|$, $\lceil K_s N((s,h),a)^\beta \rceil \geq |C((s,h),a)|$, where $K_a, K_s, \alpha, \beta$ are all constant hyperparameters. When the progressive widening condition for sampling the next state is not satisfied, either the least visited node in $C((s,h),a)$ can be selected (following PUCT), or a random node can be sampled from $C((s,h),a)$ following a distribution proportional to the number of visits (following UCT-DPW). As shown in Algorithm 1, we follow the designs of PUCT, but keep $\alpha$ and $\beta$ as constants for simplicity in hyperparameter fine-tuning.

Finally, the condition for continuing the simulation procedure, i.e., $N((s,h)) > 1$, could potentially be replaced with $N((s,h),a) > 1$ or $N((s',hars')) > 0$. These conditions indicate that the nodes $(s,h)$, $(s,h,a)$, and $(s',hars')$ have been visited before, respectively. At the end of the simulation procedure, we can either apply rollouts, i.e., simulating a single path until the end of an episode, to estimate the expected value for a leaf node $(s,h)$, or directly use $V((s,h))$ as the estimation. The former approach is widely used in online planning algorithms (Guez et al. (2013); Sunberg & Kochenderfer (2018); Lee et al. (2020)), while the latter is used in iterative frameworks like MuZero.

## C  KEY HYPERPARAMETER SETUP

| Data Type | Environment | BA-MBRL | | | | BA-MCTS | | | | BA-MCTS-SL | | | |
|---|---|---|---|---|---|---|---|---|---|---|---|---|---|
| | | $K$ | $\lambda$ | $H$ | $N$ | $K$ | $\lambda$ | $H$ | $N$ | $K$ | $\lambda$ | $H$ | $N$ |
| random | HalfCheetah | 10 | 7 | 6 | 200 | 10 | 7 | 6 | 800 | 10 | 7 | 6 | 500 |
| random | Hopper | 6 | 50 | 47 | 700 | 6 | 50 | 47 | 800 | 6 | 50 | 47 | 500 |
| random | Walker2d | 10 | 0.5 | 20 | 700 | 10 | 0.5 | 20 | 800 | 10 | 0.5 | 20 | 500 |
| medium | HalfCheetah | 12 | 6 | 6 | 300 | 12 | 6 | 5 | 800 | 12 | 6 | 5 | 500 |
| medium | Hopper | 12 | 40 | 42 | 200 | 12 | 40 | 42 | 800 | 12 | 40 | 42 | 200 |
| medium | Walker2d | 8 | 5 | 20 | 700 | 8 | 5 | 20 | 800 | 8 | 5 | 20 | 500 |
| med-replay | HalfCheetah | 11 | 40 | 10 | 300 | 11 | 40 | 10 | 800 | 11 | 40 | 10 | 500 |
| med-replay | Hopper | 7 | 5 | 5 | 700 | 7 | 5 | 5 | 800 | 7 | 5 | 5 | 500 |
| med-replay | Walker2d | 13 | 2.5 | 47 | 1000 | 13 | 2.5 | 47 | 800 | 13 | 2.5 | 47 | 500 |
| med-expert | HalfCheetah | 7 | 100 | 5 | 1000 | 7 | 100 | 5 | 800 | 7 | 100 | 5 | 1100 |
| med-expert | Hopper | 12 | 40 | 43 | 600 | 12 | 40 | 43 | 800 | 12 | 40 | 43 | 500 |
| med-expert | Walker2d | 6 | 20 | 37 | 400 | 6 | 20 | 37 | 800 | 6 | 20 | 37 | 500 |

Table 3: Key hyperparameters of the proposed algorithms for each evaluation task. $K$: ensemble size, $\lambda$: reward penalty coefficient, $H$: rollout horizon, $N$: number of training epochs.

In Table 3, we list the key hyperparameters of the proposed algorithms. For each task, an ensemble of $K$ dynamics and reward models is trained using the provided offline dataset. These learned models are then utilized as a simulator to train a control policy using off-the-shelf RL methods, such as SAC. The policy is trained for $N$ epochs. At each epoch, $50000H$ transitions are sampled by interacting with the simulator, followed by 1000 RL training iterations. In particular, 50000 states are randomly sampled from the offline dataset, with each state followed by a rollout lasting $H$ time steps. To mitigate overestimation, a reward penalty based on the discrepancy among the ensemble members is applied with a coefficient $\lambda$, as shown in Equation (5). The setups for $K$, $\lambda$, and $H$ are almost the same across the three algorithms and primarily inherited from the baseline – "Optimized" (Lu et al. (2022)), to make sure the improvements are brought by the Bayesian RL and deep search components.

The policy is evaluated on the ground truth environment for 10 episodes at the end of each training epoch. We report the average scores across the final 10 training epochs of our algorithms in Tables 1 and 5. It is important to note that increasing the number of training epochs $N$ does not necessarily lead to better policy performance, since the training is based on learned dynamics and reward models rather than the ground truth. According to (Lu et al. (2022)) and our experiments, the hyperparameters listed above can significantly influence the performance of model-based RL. Adjusting these hyperparameters could either enhance or impair the learning performance of our algorithms. We also suspect that the performance of the baselines listed in Tables 1 and 5, which are from their original papers, could be further improved by fine-tuning the relevant hyperparameters.

| Data Type | Environment | BA-MCTS | | | | | | BA-MCTS-SL | | | | | | | |
|---|---|---|---|---|---|---|---|---|---|---|---|---|---|---|---|
| | | $\rho$ | $\alpha$ | $c$ | $\eta$ | $n_s$ | $n_a$ | $\rho$ | $\alpha$ | $c$ | $\eta$ | $n_s$ | $n_a$ | $N_{SL}$ | $N_P$ |
| random | HalfCheetah | 0.1 | 0.5 | 2.5 | 0.3 | 1 | 20 | 0.1 | 0.5 | 2.5 | 0.3 | 1 | 20 | 5 | 0 |
| random | Hopper | 0.1 | 0.5 | 2.5 | 0.3 | 1 | 20 | 0.1 | 0.5 | 2.5 | 0.3 | 1 | 20 | 5 | 0 |
| random | Walker2d | 0.1 | 0.5 | 2.5 | 0.3 | 1 | 20 | 0.1 | 0.5 | 2.5 | 0.3 | 1 | 20 | 5 | 100 |
| medium | HalfCheetah | 0.1 | 0.5 | 1.0 | 0.1 | 5 | 10 | 0.1 | 0.5 | 1.0 | 0.1 | 5 | 10 | 20 | 0 |
| medium | Hopper | 0.1 | 0.5 | 2.5 | 0.3 | 1 | 20 | 0.1 | 0.5 | 1.0 | 0.3 | 1 | 20 | 5 | 0 |
| medium | Walker2d | 0.1 | 0.5 | 2.5 | 0.3 | 1 | 20 | 0.1 | 0.5 | 2.5 | 0.3 | 1 | 20 | 5 | 100 |
| med-replay | HalfCheetah | 0.1 | 0.8 | 1.0 | 0.3 | 5 | 10 | 0.1 | 0.8 | 2.5 | 0.3 | 1 | 20 | 5 | 0 |
| med-replay | Hopper | 0.1 | 0.8 | 1.0 | 0.1 | 1 | 20 | 0.1 | 0.8 | 1.0 | 0.1 | 1 | 20 | 15 | 0 |
| med-replay | Walker2d | 0.1 | 0.8 | 2.5 | 0.3 | 1 | 20 | 0.1 | 0.8 | 2.5 | 0.3 | 1 | 20 | 5 | 200 |
| med-expert | HalfCheetah | 0.1 | 0.8 | 1.0 | 0.3 | 5 | 10 | 0.1 | 0.8 | 2.5 | 0.3 | 1 | 20 | 5 | 0 |
| med-expert | Hopper | 0.1 | 0.8 | 1.0 | 0.3 | 1 | 20 | 0.1 | 0.8 | 1.0 | 0.3 | 1 | 20 | 5 | 0 |
| med-expert | Walker2d | 0.1 | 0.8 | 2.5 | 0.3 | 1 | 20 | 0.1 | 0.8 | 2.5 | 0.3 | 1 | 20 | 15 | 100 |

Table 4: Important hyperparameters used in the search process.

BA-MCTS and BA-MCTS-SL utilize Bayes Adaptive Monte Carlo Tree Search to collect samples for offline model-based RL. Instead of performing a tree search at every state, we randomly select a proportion (i.e., $\rho$) of states from the available 50000 states at each rollout time step as root nodes for tree search. For the remaining states, actions are sampled directly from the policy, i.e., $a \sim \pi(\cdot|s)$. The tree search procedure is detailed in Algorithm 1, with the number of MCTS iterations, $E$, set to 50. Increasing $\rho$ and $E$ can potentially enhance performance, but it will also linearly increase the computational cost. Table 4 outlines the key hyperparameters related to the

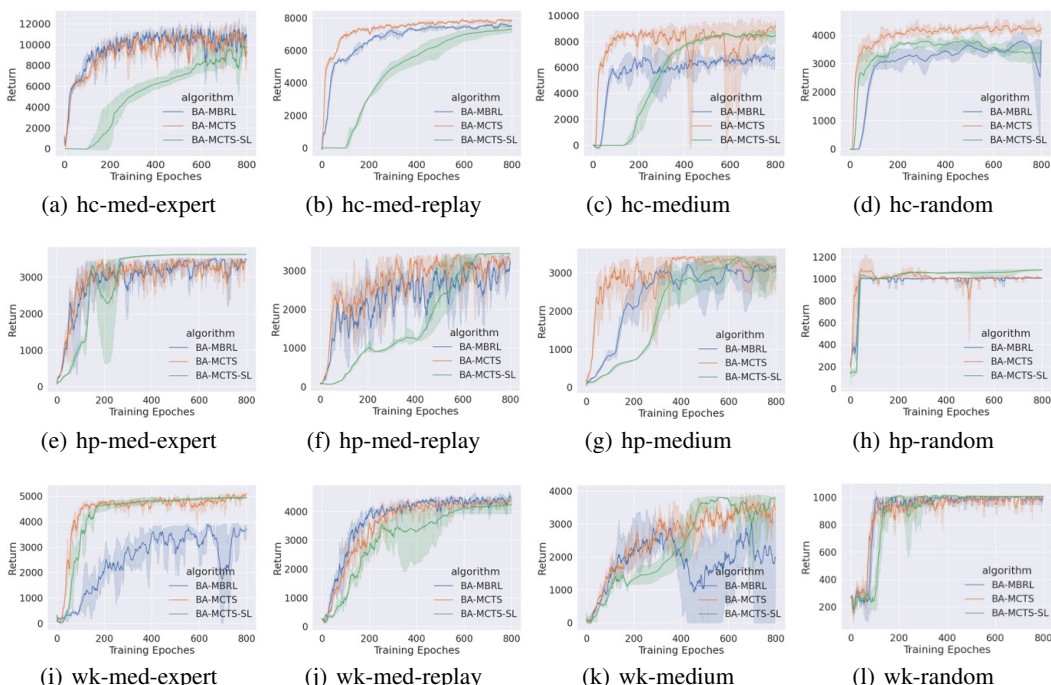

Figure 2: Performance of our proposed algorithms on D4RL MuJoCo tasks. The results for HalfCheetah, Hopper, and Walker2d are presented in the three rows, respectively. Each subfigure depicts the change in the undiscounted episodic return as a function of training epochs. Experiments are repeated three times with different random seeds, with the solid line representing the mean and the shaded area indicating the 95% confidence interval. For reference, the expert-level episodic returns for HalfCheetah, Hopper, and Walker2d are 12135, 3234.3, and 4592.3, respectively. Note that the training epochs for each algorithm, as listed in Table 3, have been linearly scaled to 800 for better visualization.

search process for each algorithm and task. (1) As described in Algorithm 1, the parameters $\alpha$ and $\beta$ control the rate of double progressive widening. ($\beta$ is set as 0.5 across all tasks.) To encourage deeper search, we limit the number of actions sampled from a state under $n_a$ and the number of next states sampled from an action under $n_s$, respectively. Action selection follows the UCT rule, as discussed in Appendix A, where $c > 0$ balances the exploration and exploitation. Additionally, inspired by the success of MuZero in enhancing exploration, we introduce Dirichlet noise $x_d$ at the root nodes, where actions are sampled from a mixture of distributions: $a \sim \eta x_d + (1 - \eta)\pi(\cdot|s)$ and $\eta$ controls the mixture rate. Notably, for $(c, \eta, n_a, n_s)$, we explore the set of possible combinations: $\{(2.5, 0.3, 20, 1), (1.0, 0.3, 20, 1), (1.0, 0.1, 20, 1), (1.0, 0.3, 10, 5), (1.0, 0.1, 10, 5)\}$ during hyperparameter fine-tuning. We believe there are likely more optimal search settings yet to be discovered. (2) In BA-MCTS-SL, policy improvement is achieved through supervised learning. We find that, rather than learning solely from samples collected within the current epoch, incorporating a buffer of samples from the past $N_{SL}$ epochs helps to stabilize the learning process. Also, in the Walker2d environment, BA-MCTS-SL requires a warm-up training phase of $N_P$ epochs using BA-MBRL, allowing the initial policy to generate effective signals for supervised learning.

## D    COMPARISONS WITH MODEL-FREE METHODS ON D4RL MUJOCO

As a complement to Table 1, we compare our algorithms with a series of model-free offline policy learning (Chen et al. (2024)) methods. We include SOTA model-free offline RL methods: CQL (Kumar et al. (2020)), BEAR (Kumar et al. (2019)), and BRAC-v (Wu et al. (2019)). Additionally, we show the performance of directly applying SAC or Behavioral Cloning (BC, Chen et al. (2024)) to the provided offline dataset in the last two columns. The mean performance of the baselines are

| Data Type | Environment | BA-MCTS -SL (ours) | BA-MCTS (ours) | BA-MBRL (ours) | CQL | BEAR | BRAC-v | SAC | BC |
|---|---|---|---|---|---|---|---|---|---|
| random | HalfCheetah | $29.20 \pm 2.00$ | $\mathbf{36.23} \pm 1.04$ | $32.76 \pm 1.16$ | 35.4 | 25.1 | 31.2 | 30.5 | 2.1 |
| random | Hopper | $\mathbf{33.83} \pm 0.10$ | $31.56 \pm 0.12$ | $31.47 \pm 0.03$ | 10.8 | 11.4 | 12.2 | 11.3 | 1.6 |
| random | Walker2d | $\mathbf{21.89} \pm 0.07$ | $21.59 \pm 0.32$ | $21.45 \pm 0.53$ | 7.0 | 7.3 | 1.9 | 4.1 | 9.8 |
| medium | HalfCheetah | $70.47 \pm 3.52$ | $\mathbf{75.84} \pm 3.81$ | $56.54 \pm 5.20$ | 44.4 | 41.7 | 46.3 | -4.3 | 36.1 |
| medium | Hopper | $97.75 \pm 7.09$ | $96.70 \pm 14.0$ | $\mathbf{98.25} \pm 3.42$ | 86.6 | 52.1 | 31.1 | 0.8 | 29.0 |
| medium | Walker2d | $\mathbf{82.24} \pm 1.85$ | $74.73 \pm 3.25$ | $75.41 \pm 4.17$ | 74.5 | 59.1 | 81.1 | 0.9 | 6.6 |
| med-replay | HalfCheetah | $61.16 \pm 1.60$ | $\mathbf{65.45} \pm 0.81$ | $62.50 \pm 0.18$ | 46.2 | 38.6 | 47.7 | -2.4 | 38.4 |
| med-replay | Hopper | $\mathbf{106.3} \pm 0.13$ | $101.8 \pm 3.46$ | $93.91 \pm 4.25$ | 48.6 | 33.7 | 0.6 | 3.5 | 11.8 |
| med-replay | Walker2d | $92.13 \pm 5.13$ | $95.06 \pm 2.11$ | $\mathbf{97.54} \pm 1.93$ | 32.6 | 19.2 | 0.9 | 1.9 | 11.3 |
| med-expert | HalfCheetah | $80.53 \pm 6.63$ | $76.16 \pm 10.3$ | $\mathbf{90.52} \pm 4.13$ | 62.4 | 53.4 | 41.9 | 1.8 | 35.8 |
| med-expert | Hopper | $\mathbf{112.2} \pm 0.29$ | $108.3 \pm 0.22$ | $107.8 \pm 0.37$ | 111 | 96.3 | 0.8 | 1.6 | 111.9 |
| med-expert | Walker2d | $107.7 \pm 0.82$ | $\mathbf{110.0} \pm 1.74$ | $84.71 \pm 0.87$ | 98.7 | 40.1 | 81.6 | -0.1 | 6.4 |
| Average Score | | $\mathbf{74.62}$ | 74.45 | 71.06 | 54.85 | 39.83 | 31.44 | 4.13 | 25.07 |

Table 5: Comparisons between the proposed algorithms and model-free offline policy learning methods on the D4RL benchmark suite. Each value represents the normalized score, as proposed in (Fu et al. (2020)), of the policy trained by the corresponding algorithm. These scores are undiscounted returns normalized to approximately range between 0 and 100, where a score of 0 corresponds to a random policy and a score of 100 corresponds to an expert-level policy. For our algorithms, we report the average score of the final ten policy learning epochs and its standard deviation across three different random seeds.

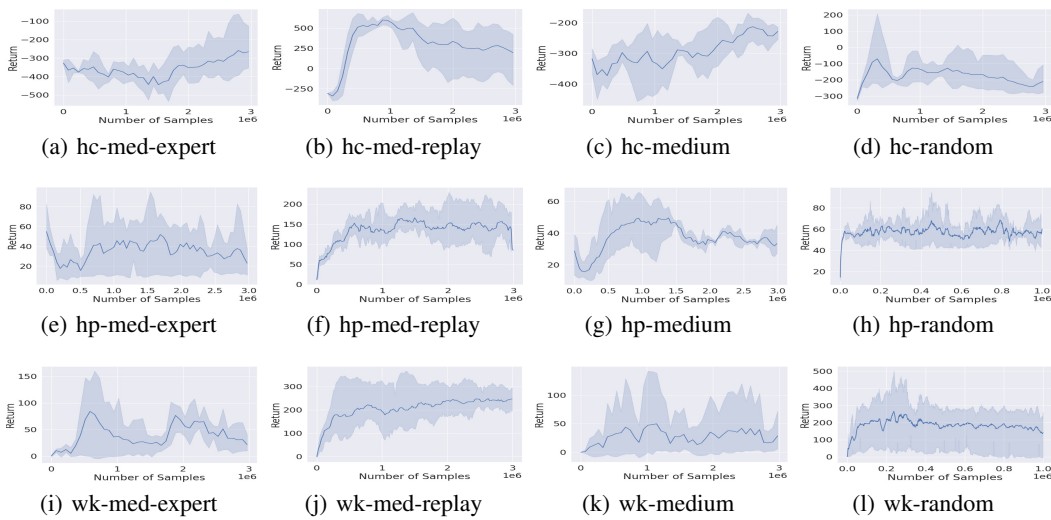

(a) hc-med-expert    (b) hc-med-replay    (c) hc-medium    (d) hc-random

(e) hp-med-expert    (f) hp-med-replay    (g) hp-medium    (h) hp-random

(i) wk-med-expert    (j) wk-med-replay    (k) wk-medium    (l) wk-random

Figure 3: Performance of Sampled EfficientZero on D4RL MuJoCo tasks. The results for HalfCheetah, Hopper, and Walker2d are presented in the three rows, respectively. Each subfigure depicts the change in undiscounted episodic return as a function of the number of training samples. Experiments are repeated three times with different random seeds, with the solid line representing the mean and the shaded area indicating the 95% confidence interval. For reference, the expert-level episodic returns for HalfCheetah, Hopper, and Walker2d are 12135, 3234.3, and 4592.3, respectively.

taken from related works (Yu et al. (2020); Kidambi et al. (2020); Fu et al. (2020)). Our algorithms show significantly better performance, demonstrating the necessity of model-based learning in these environments. The training plots of our proposed algorithms in each environment is further detailed in Figure 2.

# E   COMPUTATION COST ON D4RL MUJOCO

In Table 6, we report the training time of the proposed algorithm and Sampled EfficientZero on the D4RL MuJoCo tasks. The experiments were conducted on a server with 40 Intel(R) Xeon(R) Gold 5215 CPUs and 4 Tesla V100-SXM2-32GB GPUs. While the tree-search-based variants (i.e., BA-MCTS and BA-MCTS-SL) achieve higher performance, they require more computation during

the offline training stage. However, this extra computational cost is limited to the training phase; no MCTS is performed during deployment, to ensure real-time execution. Additionally, leveraging parallel computation frameworks for MCTS could further reduce the training time. On the other hand, our algorithm requires considerably less training time than Sampled EfficientZero, which is also based on deep search, to achieve superior performance, as shown in Figures 3 and 2.

| Data Type | Environment | BA-MCTS -SL (ours) | BA-MCTS (ours) | BA-MBRL (ours) | Sampled EfficientZero |
|---|---|---|---|---|---|
| random | HalfCheetah | $11.2 \pm 2.6$ | $14.6 \pm 2.2$ | $1.2 \pm 0.4$ | $54.8 \pm 2.6$ |
| random | Hopper | $48.7 \pm 2.4$ | $65.2 \pm 1.9$ | $5.6 \pm 0.8$ | $153.7 \pm 19.4$ |
| random | Walker2d | $25.7 \pm 2.0$ | $38.1 \pm 1.1$ | $5.2 \pm 1.1$ | $175.7 \pm 28.2$ |
| medium | HalfCheetah | $12.6 \pm 3.0$ | $15.3 \pm 1.3$ | $1.9 \pm 0.2$ | $54.7 \pm 1.9$ |
| medium | Hopper | $18.9 \pm 4.8$ | $67.5 \pm 0.2$ | $1.8 \pm 0.2$ | $70.5 \pm 1.0$ |
| medium | Walker2d | $24.0 \pm 1.8$ | $33.3 \pm 4.0$ | $4.8 \pm 1.0$ | $63.7 \pm 1.5$ |
| med-replay | HalfCheetah | $24.7 \pm 0.9$ | $22.4 \pm 1.2$ | $2.1 \pm 0.4$ | $54.8 \pm 1.8$ |
| med-replay | Hopper | $17.9 \pm 0.0$ | $12.7 \pm 1.2$ | $5.4 \pm 0.2$ | $126.6 \pm 12.0$ |
| med-replay | Walker2d | $40.3 \pm 7.2$ | $77.0 \pm 2.4$ | $8.0 \pm 0.3$ | $134.9 \pm 4.2$ |
| med-expert | HalfCheetah | $32.6 \pm 0.1$ | $12.6 \pm 0.8$ | $4.7 \pm 1.1$ | $55.4 \pm 1.3$ |
| med-expert | Hopper | $61.5 \pm 4.6$ | $76.9 \pm 11.6$ | $5.3 \pm 0.5$ | $66.1 \pm 0.9$ |
| med-expert | Walker2d | $35.0 \pm 6.5$ | $55.9 \pm 1.8$ | $2.9 \pm 0.9$ | $60.8 \pm 1.8$ |

Table 6: Training time (in hours) of the proposed algorithms and Sampled EfficientZero for each evaluation task. Results are presented as the mean and standard deviation from three repeated experiments.

## F  ABLATION STUDY ON THE REWARD PENALTY

| Data Type | Environment | BA-MCTS -SL | BA-MCTS | BA-MBRL | BA-MCTS -SL ($\lambda = 0$) | BA-MCTS ($\lambda = 0$) | BA-MBRL ($\lambda = 0$) |
|---|---|---|---|---|---|---|---|
| random | HalfCheetah | $29.20 \pm 2.00$ | $36.23 \pm 1.04$ | $32.76 \pm 1.16$ | $34.80 \pm 1.39$ | $38.78 \pm 1.65$ | $\mathbf{39.64} \pm 2.86$ |
| random | Hopper | $\mathbf{33.83} \pm 0.10$ | $31.56 \pm 0.12$ | $31.47 \pm 0.03$ | $9.16 \pm 0.16$ | $7.44 \pm 0.14$ | $6.97 \pm 0.07$ |
| random | Walker2d | $\mathbf{21.89} \pm 0.07$ | $21.59 \pm 0.32$ | $21.45 \pm 0.53$ | $17.53 \pm 6.16$ | $21.53 \pm 0.42$ | $21.41 \pm 0.64$ |
| medium | HalfCheetah | $70.47 \pm 3.52$ | $\mathbf{75.84} \pm 3.81$ | $56.54 \pm 5.20$ | $61.64 \pm 4.58$ | $60.84 \pm 2.00$ | $41.49 \pm 2.29$ |
| medium | Hopper | $97.75 \pm 7.09$ | $96.70 \pm 14.0$ | $98.25 \pm 3.42$ | $102.8 \pm 2.29$ | $\mathbf{104.4} \pm 1.88$ | $93.68 \pm 11.4$ |
| medium | Walker2d | $82.24 \pm 1.85$ | $74.73 \pm 3.25$ | $75.41 \pm 4.17$ | $\mathbf{82.61} \pm 0.86$ | $57.01 \pm 7.24$ | $57.97 \pm 15.4$ |
| med-replay | HalfCheetah | $61.16 \pm 1.60$ | $\mathbf{65.45} \pm 0.81$ | $62.50 \pm 0.18$ | $42.10 \pm 2.85$ | $36.65 \pm 2.39$ | $44.03 \pm 7.35$ |
| med-replay | Hopper | $106.3 \pm 0.13$ | $101.8 \pm 3.46$ | $93.91 \pm 4.25$ | $\mathbf{107.9} \pm 0.07$ | $84.11 \pm 2.97$ | $91.81 \pm 11.5$ |
| med-replay | Walker2d | $92.13 \pm 5.13$ | $95.06 \pm 2.11$ | $97.54 \pm 1.93$ | $88.61 \pm 5.21$ | $97.33 \pm 3.51$ | $\mathbf{98.19} \pm 1.23$ |
| med-expert | HalfCheetah | $80.53 \pm 6.63$ | $76.16 \pm 10.3$ | $\mathbf{90.52} \pm 4.13$ | $51.76 \pm 5.31$ | $26.60 \pm 1.46$ | $29.88 \pm 2.28$ |
| med-expert | Hopper | $\mathbf{112.2} \pm 0.29$ | $108.3 \pm 0.22$ | $107.8 \pm 0.37$ | $106.8 \pm 6.34$ | $81.76 \pm 6.45$ | $86.79 \pm 18.7$ |
| med-expert | Walker2d | $107.7 \pm 0.82$ | $110.0 \pm 1.74$ | $84.71 \pm 0.87$ | $\mathbf{110.8} \pm 1.72$ | $110.2 \pm 0.91$ | $53.35 \pm 38.0$ |
| Average Score | | $\mathbf{74.62}$ | $74.45$ | $71.06$ | $68.04$ | $60.55$ | $55.43$ |

Table 7: Comparison of the proposed algorithms with their corresponding versions without the reward penalty (i.e., $\lambda = 0$). The definitions of the values in this table are consistent with those in Table 5.

To demonstrate the necessity of incorporating the reward penalty in offline MBRL, we conduct an ablation study by setting $\lambda$ in Eq. (5) to 0, resulting in ablated versions of our proposed three algorithms. The results are presented in Table 7. **First**, the average performance of the algorithms with the reward penalty is consistently better, demonstrating the importance of using reward penalties in offline MBRL to prevent the overexploitation of the learned world models (which can be inaccurate). **Second**, the supervised-learning-based algorithm (i.e., BA-MCTS-SL ($\lambda = 0$)) is less affected by the absence of the reward penalty, compared to the policy-gradient-based methods. Notably, BA-MCTS-SL ($\lambda = 0$) and BA-MCTS-SL achieve comparable performance in the Hopper and Walker2d tasks. This shows an additional advantage of BA-MCTS-SL – its reduced sensitivity to model inaccuracies. **Lastly**, there are instances where superior performance is achieved with $\lambda$ set to 0. For example, BA-MCTS-SL ($\lambda = 0$) performs better than BA-MCTS-SL in 5 out of 12 tasks. This suggests that the performance of our algorithms in Tables 1 and 5 could be further improved by adjusting hyperparameters such as $\lambda$[9].

---

[9] As mentioned in Section 5, we retain most of the hyperparameter settings from Optimized (Lu et al. (2022)) to ensure that the performance improvements are attributed to our algorithm design.

# G    DETAILS OF THE TOKAMAK CONTROL TASKS

| STATE SPACE | |
|---|---|
| Scalar States | $\beta_n$, Internal Inductance, Line Averaged Density, Loop Voltage, Stored Energy |
| Profile States | Electron Density, Electron Temperature, Pressure, Safety Factor, Ion Temperature, Ion Rotation |

| ACTION SPACE | |
|---|---|
| Targets | Current Target, Density Target |
| Shape Variables | Elongation, Top Triangularity, Bottom Triangularity, Minor Radius, Radius and Vertical Locations of the Plasma Center |
| Direct Actuators | Power Injected, Torque Injected, Total Deuterium Gas Injection, Total ECH Power, Magnitude and Sign of the Toroidal Magnetic Field |

Table 8: The state and action spaces of the tokamak control tasks.

Nuclear fusion is a promising energy source to meet the world's growing demand. It involves fusing the nuclei of two light atoms, such as hydrogen, to form a heavier nucleus, typically helium, releasing energy in the process. The primary challenge of fusion is confining a plasma, i.e., an ionized gas of hydrogen isotopes, while heating it and increasing its pressure to initiate and sustain fusion reactions. The tokamak is one of the most promising confinement devices. It uses magnetic fields acting on hydrogen atoms that have been ionized (given a charge) so that the magnetic fields can exert a force on the moving particles (Pironti & Walker (2005)).

Char et al. (2024) trained a deep recurrent network as a dynamics model for the DIII-D tokamak, a device located in San Diego, California, and operated by General Atomics, using a large dataset of operational data from that device. A typical shot (i.e., episode) on DIII-D lasts around 6-8 seconds, consisting of a one-second ramp-up phase, a multi-second flat-top phase, and a one-second ramp-down phase. The DIII-D also features several real-time and post-shot diagnostics that measure the magnetic equilibrium and plasma parameters with high temporal resolution. The authors demonstrate that the learned model predicts these measurements for entire shots with remarkable accuracy. Thus, we use this model as a "ground truth" simulator for tokamak control tasks. Specifically, we generate a dataset of 725270 transitions for offline RL and evaluate the learned policy using this data-driven simulator.

The state and action spaces for the tokamak control tasks are outlined in Table 8. For detailed physical explanations of their components, please refer to (Abbate et al. (2021); Char et al. (2023); Ariola et al. (2008)). The state space consists of five scalar values and six profiles which are discretized measurements of physical quantities along the minor radius of the toroid. After applying principal component analysis (Maćkiewicz & Ratajczak (1993)), the pressure profile is reduced to two dimensions, while the other profiles are reduced to four dimensions each. In total, the state space comprises 27 dimensions. The action space includes direct control actuators for neutral beam power, torque, gas, ECH power, current, and magnetic field, as well as target values for plasma density and plasma shape, which are managed through a lower-level control module. Altogether, the action space consists of 14 dimensions. While for certain tasks, it is possible to prune the state and action spaces to reduce the learning complexity, we have chosen not to apply any domain-specific knowledge in these evaluations for general RL algorithms. We reserve the domain-specific applications of our algorithms, which would require more domain knowledge and engineering efforts, as an important future work.

We select a reference shot from DIII-D, which spans 251 time steps, and use its trajectories of Ion Rotation, Electron Temperature, and $\beta_n$ as targets for three tracking tasks. Specifically, $\beta_n$ is the normalized ratio between plasma pressure and magnetic pressure, a key quantity serving as a rough economic indicator of efficiency. Since the tracking targets vary over time, we include the time step as part of the policy input. The reward function for each task is defined as the negative squared tracking error of the corresponding component (i.e., temperature, rotation, or $\beta_n$) at each time step, and the reward is normalized by the episode horizon (i.e., 251 time steps). Notably, for policy learning, the reward function is provided rather than learned from the offline dataset as in D4RL tasks; and the dataset does not include the reference shot or any nearby, similar shots.

## H  ANALYSIS OF THE BENEFITS OF BELIEF ADAPTATION

| hc-med-expert | hc-med-replay | hc-medium | hc-random |
|---|---|---|---|
| 0.7515 | 2.6315 | 2.2674 | 2.4561 |

| hp-med-expert | hp-med-replay | hp-medium | hp-random |
|---|---|---|---|
| 14.064 | 4.2214 | 3.5913 | 1.7400 |

| wk-med-expert | wk-med-replay | wk-medium | wk-random |
|---|---|---|---|
| 2.0898 | 3.8169 | 34.264 | 1.0341 |

Table 9: Ratios of average transition likelihoods in offline data with and without belief adaptation across ensemble members. Bayesian belief adaptation based on observed transitions generally enhances prediction performance in the offline dataset, with the exception of hc-med-expert (i.e., the HalfCheetah dataset with medium-expert performance).

| Data Type | Environment | Prediction Error on Next State | | Prediction Error on Reward | | Overall Prediction Error | |
|---|---|---|---|---|---|---|---|
| | | Adaptive | Uniform | Adaptive | Uniform | Adaptive | Uniform |
| random | HalfCheetah | **0.339** ± .021 | 0.342 ± .017 | **0.016** ± .001 | **0.016** ± .001 | **0.177** ± .011 | 0.179 ± .009 |
| random | Hopper | 0.232 ± .016 | **0.189** ± .008 | **0.012** ± .001 | 0.022 ± .008 | 0.122 ± .008 | **0.106** ± .008 |
| random | Walker2d | **62.10** ± 11.0 | 112.0 ± 19.0 | **1.364** ± .215 | 4.908 ± .708 | **31.73** ± 5.61 | 58.44 ± 9.83 |
| medium | HalfCheetah | 1.387 ± .040 | **1.355** ± .038 | **0.057** ± .001 | 0.061 ± .001 | 0.722 ± .021 | **0.708** ± .019 |
| medium | Hopper | **0.429** ± .033 | 0.570 ± .035 | **19.03** ± 8.58 | 68.24 ± 11.4 | **9.727** ± 4.30 | 34.40 ± 5.69 |
| medium | Walker2d | **34.01** ± .556 | 34.26 ± .142 | **24.38** ± 4.31 | 113.1 ± 3.47 | **29.19** ± 1.89 | 73.69 ± 1.78 |
| med-replay | HalfCheetah | **0.677** ± .027 | 0.707 ± .036 | 0.115 ± .005 | **0.114** ± .006 | **0.396** ± .016 | 0.410 ± .021 |
| med-replay | Hopper | **0.170** ± .022 | 0.212 ± .054 | **1.177** ± .420 | 3.320 ± 1.14 | **0.674** ± .221 | 1.766 ± .558 |
| med-replay | Walker2d | 66.83 ± 7.39 | **44.61** ± 1.34 | **27.21** ± 3.24 | 60.52 ± 3.17 | **47.02** ± 5.17 | 52.57 ± 2.25 |
| med-expert | HalfCheetah | **1.423** ± .054 | 1.467 ± .046 | **0.071** ± .002 | 0.074 ± .002 | **0.747** ± .028 | 0.771 ± .024 |
| med-expert | Hopper | **3.665** ± 2.56 | 18.27 ± 2.09 | **71.46** ± 79.9 | 384.7 ± 37.6 | **37.56** ± 41.2 | 201.5 ± 19.4 |
| med-expert | Walker2d | 55.69 ± 3.71 | **51.89** ± .481 | **121.9** ± 12.6 | 186.3 ± 3.86 | **88.77** ± 8.18 | 119.1 ± 2.08 |

Table 10: Comparison of the prediction errors for next states and rewards in imaginary rollouts, with and without Bayesian belief adaptation. The last two columns present the average prediction errors for next states and rewards, serving as an overall indicator. Each metric is computed over three repetitions with different random seeds, reporting both the mean and standard deviation. Ground truth for the imaginary rollouts is obtained by replaying the action sequences in the real simulators.

The Bayesian adaptation, as defined in Eq. (4), uses observed state transition sequences to adjust the belief over each ensemble member, thereby enhancing the quality of predicted rollout trajectories. For each benchmarking task in D4RL MuJoCo, we compute the average transition likelihood (i.e., $\mathcal{P}(s'|s,a)\mathcal{R}(r|s,a)$) through the provided offline rollouts, comparing cases with and without belief adaptation. The ratios of these average likelihoods are presented in Table 9. Results show that the offline rollouts, collected from real environments, are more likely under the adapted ensemble, with the exception of HalfCheetah-med-expert. This also demonstrates that Bayesian belief adaptation serves as an effective calibration method for learned dynamics models.

As detailed in Appendix C, at each learning epoch, we randomly sample states from the offline dataset to serve as starting points for imaginary rollouts, which are generated using the learned dynamics models. Belief adaptation is applied not only to the offline trajectories (to establish beliefs at these starting states) but also throughout the imaginary rollouts. To evaluate the quality of the imaginary rollouts, we select 10000 starting states from the offline dataset and perform rollouts using the same ensemble and behavior policy (based on MCTS), with the only difference being whether belief adaptation is applied to the ensemble members. Table 10 presents the prediction errors, measured as the average mean squared errors across each time step in the rollouts, for next states and rewards in the imaginary rollouts. Ground truth rollouts are generated by replaying the planned action sequences in the real simulators. The last two columns, which show the average prediction errors for next states and rewards, reveal that the adaptive ensemble achieves more accurate predictions in 10 out of 12 benchmarking tasks, with comparable performance in the remaining two.

Regarding the overall prediction error, the adaptive ensemble significantly outperforms the uniform ensemble in Hopper-med-expert. To illustrate this, we plot the belief adaptation over an offline trajectory in Hopper-med-expert in Figure 4(a). Initially, all twelve ensemble members have the same belief. As the trajectory progresses, the beliefs of each member are updated based on the transition history, with the dominant model (the one with the highest belief) continuously changing.

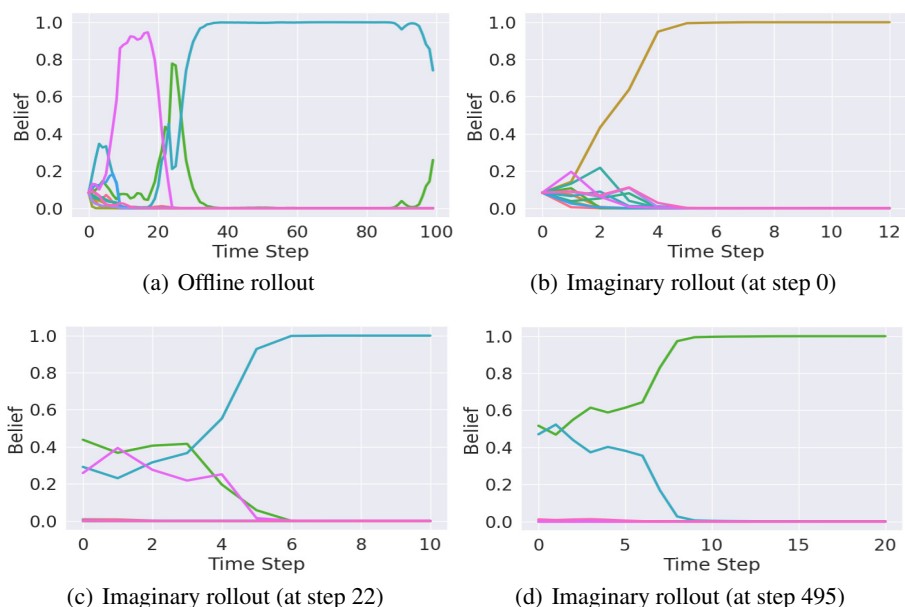

(a) Offline rollout

(b) Imaginary rollout (at step 0)

(c) Imaginary rollout (at step 22)

(d) Imaginary rollout (at step 495)

Figure 4: Belief adaptation during offline and imaginary rollouts. (a) shows the belief over twelve ensemble members, each represented by a specific color, adapting to an offline trajectory of Hopper-med-expert. (b), (c), and (d) illustrate the belief changes during imaginary rollouts which start from the beginning, middle, and end of the offline trajectory shown in (a), respectively.

Additionally, we track the belief changes in imaginary rollouts starting from the beginning, middle, and end of the offline trajectory. It is evident that different ensemble members dominate at different stages of the offline rollout. This dynamic belief adaptation is crucial for achieving lower prediction errors as shown in Table 10.

# I COMPARISON WITH ADDITIONAL BASELINES

| Data Type | Environment | BA-MCTS (ours) | MOBILE | CBOP | RAMBO | APE-V | MAPLE |
|---|---|---|---|---|---|---|---|
| random | HalfCheetah | 39.09* $\pm$ 1.30 | 39.3 | 32.8 | 40.0 | 29.9 | **41.5** |
| random | Hopper | 31.56 $\pm$ 0.12 | **31.9** | 31.4 | 21.6 | 31.3 | 10.7 |
| random | Walker2d | 21.59 $\pm$ 0.32 | 17.9 | 17.8 | 11.5 | 15.5 | **22.1** |
| medium | HalfCheetah | 75.84 $\pm$ 3.81 | 74.6 | 74.3 | **77.6** | 69.1 | 48.5 |
| medium | Hopper | 103.9* $\pm$ 0.33 | **106.6** | 102.6 | 92.8 | - | 44.1 |
| medium | Walker2d | 87.25* $\pm$ 2.64 | 87.7 | **95.5** | 86.9 | 90.3 | 81.3 |
| med-replay | HalfCheetah | 70.16* $\pm$ 5.24 | **71.7** | 66.4 | 68.9 | 64.6 | 69.5 |
| med-replay | Hopper | **106.4*** $\pm$ 0.53 | 103.9 | 104.3 | 96.6 | 98.5 | 85.0 |
| med-replay | Walker2d | **95.06** $\pm$ 2.11 | 89.9 | 92.7 | 85.0 | 82.9 | 75.4 |
| med-expert | HalfCheetah | 100.6* $\pm$ 0.87 | **108.2** | 105.4 | 93.7 | 101.4 | 55.4 |
| med-expert | Hopper | **112.8** $\pm$ 0.14 | 112.6 | 111.6 | 83.3 | 105.7 | 95.3 |
| med-expert | Walker2d | 116.0* $\pm$ 1.49 | 115.2 | **117.2** | 68.3 | 110.0 | 107.0 |
| Average Score | | **80.02** | 79.96 | 79.33 | 68.85 | 72.65 | 61.32 |

Table 11: Comparison of BA-MCTS with more recent offline RL baselines on the D4RL benchmark suite. Each value in the table represents the normalized score as defined in Tables 1 and 5. Baseline results are sourced from their respective papers: MOBILE (Sun et al. (2023)), CBOP (Jeong et al. (2023)), RAMBO (Rigter et al. (2022)), APE-V (Ghosh et al. (2022)), and MAPLE (Chen et al. (2021)). For BA-MCTS, results marked with * indicate enhanced performance achieved using the reward penalty design proposed in (Sun et al. (2023)).

As noted in Section 5, the implementation of our algorithms is based on Optimized, making minimal changes to its codebase and hyperparameter settings. Therefore, the performance improvements shown in Table 1 stem from the Bayesian RL framework and deep search component. Our algo-

rithms can be seamlessly integrated with other advancements in offline MBRL, such as RNN-based policy functions, more accurate world model learning, and improved uncertainty quantification.

To testify this, we replace the reward penalty design in Eq. (5) with the one proposed in a recent work (Sun et al. (2023)). Specifically, this reward penalty measures the discrepancy in the Q-value targets predicted by each ensemble member and is calculated based not only on the ensemble but also on the target Q network:

$$\tilde{r}(s, a, r) = r - \lambda \cdot \text{std} \left[ \frac{\gamma}{M} \sum_{j=1}^{M} Q_{\psi^-}(s'_{i,j}, a'_{i,j}) \right]_{i=1}^{K}, \ s'_{i,j} \sim \mathcal{P}_{\theta}^{i}(\cdot|s, a), \ a'_{i,j} \sim \pi(\cdot|s'_{i,j}) \quad (7)$$

Here, $(s'_{i,j}, a'_{i,j})$ are samples generated from the learned dynamics and policy models. These samples are fed into the target Q network, $Q_{\psi^-}$, to estimate the Q-value target for the current state-action pair $(s, a)$. The reward penalty is computed as the standard deviation of the estimated Q-value targets across different ensemble members.

With this modification, BA-MCTS achieves improved performance in several environments (compared to Table 1), as indicated by * in Table 11, and demonstrates state-of-the-art (SOTA) overall performance on the D4RL MuJoCo benchmark. Among the baselines in Table 11, APE-V and MAPLE employ adaptive policies implemented with RNNs[10], while RAMBO, CBOP, and MO-BILE are more recent baselines. Comparing BA-MCTS to these baselines showcases its SOTA performance. Note that for results marked with *, the same ensemble of world models and hyperparameter setup as MOBILE are used, and further fine-tuning is likely to enhance the performance of BA-MCTS.

---

[10]APE-V is a model-free algorithm and MAPLE does not leverage a Bayesian RL framework, which are different from our algorithm design.

