# OpenReview forum: "Bayes Adaptive Monte Carlo Tree Search for Offline Model-based Reinforcement Learning"
_ICLR.cc/2025/Conference — Submitted to ICLR 2025_

### Official Review · Reviewer_grWz · 2024-11-03

**Soundness:** 3
**Presentation:** 1
**Contribution:** 2
**Rating:** 5
**Confidence:** 3

**Summary:**

Model-based offline reinforcement learning is a promising approach to improve the data efficiency and the generalization capability beyond the training dataset. To address the uncertainty issue when fitting the model with a static dataset, the conventional approach is to use an ensemble of world models, from which the uncertainty can be estimated. This work focuses on the adaptive ensemble mechanism where the belief over each model in the ensemble will be updated when the new samples are experienced by the algorithms. The proposed Bayes Adaptive Monte-Carlo planning algorithm has two key contributions: cast the offline MBRL as Bayes Adaptive MDP (BAMDP), based on which a Monte Carlo tree search method is used for planning. The experiments on D4RL MuJoCo and the three target tracking tasks show that the proposed method has better performance compared with state-of-the-art model-based and model-free offline RL methods.

**Strengths:**

- Experiments. This work considers both the D4RL MuJoCo and challenging stochastic target tracking tasks. The experiments also include an ablation study where each component of the proposed algorithms is verified, i.e., BA-MBRL, BA-MCTS, BA-MCTS-SL. The results are compared with many SOTA methods such as COMBO, MoReL, MOPO.
- Problem Formulation. I find the adaptive ensemble mechanism is indeed a crucial problem in the ensemble-based method, and the proposed method using the BAMDP as a backbone to address this issue is novel. The algorithms are well-motivated.  I do believe that this paper has good contribution on the offline model based RL community.

**Weaknesses:**

- The comparison with related work is unclear. In particular, MuZero is a major  comparison object to motivate this study, while the statement in the paper is not clear. In Section 3, the paper claims that MuZero does not apply uncertainty estimation of the learned model, which has no direct relation with using latent models (i.e., estimating uncertainty in the latent space is possible). Please provide a more structured comparison of key related works to highlight the specific gaps.
- Method section is confusing. Section 4.1 titled “ key role of deep ensemble,” while the contents are more about introducing BAMDP and reward design to incorporate the pessimistic-MDP. In Section 4.2, the newly introduced $(s,h)$ as a decision point seems out of nowhere, what is the connection with $(s,b)$ as introduced in BAMDP? The introduction of the algorithm in Section 4.2 is very hard to follow, and a paragraph title can help a lot. For instance, in the last paragraph where introducing the Bayes-optimal policy,  the author should directly introduce the design choice and then compare it with the previous methods.
- Lack of comparison of the uncertainty evaluation. One of the key motivations of the proposed method is to use the samples to update the belief of ensembles. In the experiments, the related results on the benefits of such adaptation for ensemble-based methods are lacking (e.g., set the adaptive parameter to be zero).

**Questions:**

- what is the major purpose of having Fig.1?
- What is the explanation of the lack of the convergence in Figure 2 for optimized method? Why in the last subfigure all the curves seem to struggle  to converge?
- How to select $\lambda$ in the reward and the number of the ensembles in practice?

---

> ### Author Response · Authors · 2024-11-25
>
> ### For Weakness 1:
>
> In the revised manuscript, we have restructured the related work section (Section 3) by adding paragraph titles. Additionally, we provide a more structured comparison with key related works, including MuZero, Bayesian approaches to offline RL, and algorithms that leverage model-based search to enhance the efficiency of actor-critic training. These modifications are highlighted in blue for ease of reference.
>
> Our algorithm fundamentally differs from MuZero and its variants. (1) MuZero integrates model learning and policy training into a single stage, and the learned world models are defined on a latent state space rather than the real state space. In contrast, our algorithm and the baselines, such as Optimized and MOPO, first learn an ensemble of world models (until their convergence), and then use them as a surrogate simulator for policy learning.  (2) MuZero learns a single latent model as the world model and does not involve a mechanism to handle the uncertainty of dynamics/reward predictions. In contrast, our algorithm learns an ensemble of world models, dynamically adapts beliefs for each ensemble member, and constructs a pessimistic MDP to mitigate over-estimation. While estimating uncertainty in the latent space is possible, MuZero does not implement such mechanisms. (3) MuZero does not utilize double progressive widening (DPW, Auger et al. (2013)) or bayes-adaptive planning in its MCTS process, as we do in Algorithm 1. DPW can significantly enhance planning performance in environments with continuous state/action spaces and stochastic dynamics.
>
> ### For Weakness 2:
>
> As noted in Section 4.1, the practical implementation of a BAMDP relies on the use of deep ensembles, because the ensemble ​​$\{(P_{\theta}^i, R_{\theta}^i)_{i=1}^K\}$ constitute a finite approximation of the space of world models. Also, to construct a pessimistic-MDP, we introduce a reward penalty defined based on the discrepancy in predictions by the deep ensembles, as defined in Eq. (5). Both of these key designs are dependent on deep ensembles, which is why the section is titled “The Key Role of Deep Ensembles”.
>
> As mentioned in Footnote 4, the states in BAMDPs consist of both $s$ and the corresponding belief $b(\theta)$. $b(\theta)$ is a function of, and implied by, the history $h$, since $b(\theta)$ is updated in a Bayesian manner based on the history, as shown in Eq. (4).
>
> Thank you for your valuable feedback. We have refined Section 4.2 by reorganizing the content and adding paragraph titles. Additionally, we have clarified the design motivations and emphasized our novel contributions. Please check the modifications highlighted in blue. In the last paragraph, the introduction of alternative design choices and comparisons between previous methods and ours involve extensive details, so we have to move this content to Appendix B.
>
> ### For Weakness 3:
>
> In Table 1, the performance gap between BA-MBRL and Optimized is solely attributable to the use of samples to update the belief of ensembles.
>
> As mentioned in the first paragraph of Section 5.1, BA-MBRL follows existing offline MBRL methods such as Optimized, MOPO, and MOReL, by learning an ensemble of world models, applying reward penalties to construct a pessimistic MDP, and using the pessimistic MDP as a surrogate simulator for RL training. Particularly, BA-MBRL and Optimized both use the standard deviation of the predictions across all ensemble members (i.e., Eq. (5)) as the reward penalty design. Thus, the **ONLY** difference between BA-MBRL and Optimized lies in whether the belief over each ensemble member (i.e., $b(\theta)$) is adapted according to Eq. (4). The benefit of this belief adaptation is clearly demonstrated in Table 1.
>
> In Appendix H of the revised manuscript, we show that employing a Bayes-adaptive ensemble, instead of a uniform ensemble, improves the prediction likelihood for the provided offline trajectories and reduces the prediction errors in imaginary rollouts. As illustrative examples of this performance improvement, Figure 4 tracks belief adaptation during an offline rollout and several imaginary rollouts.

---

> ### Author Response · Authors · 2024-11-25
>
> ### For Question 1:
>
> As discussed in Section 5.2, MuZero and its variants also utilize MCTS in model-based RL. Specifically, both MuZero and BA-MCTS-SL (ours) rely on supervised learning for policy improvement. Therefore, MuZero and its variants are included as baselines in our comparisons. According to Niu et al. (2023), Sampled EfficientZero (Ye et al., 2021) demonstrates the best performance among MuZero variants on (online) MuJoCo locomotion tasks. Thus, we evaluate Sampled EfficientZero on the set of D4RL MuJoCo tasks, as shown in Figure 3 (which was originally Figure 1). The results indicate that our proposed algorithms outperform Sampled EfficientZero while incurring significantly lower computational costs, as detailed in Appendix E.
>
> ### For Question 2:
>
> As mentioned in Section 5.3, the tracking tasks in Tokamak control have a 28-dimensional state space and a 14-dimensional action space, both continuous. Moreover, these tasks are highly stochastic, as the underlying dynamics model is a probabilistic neural network and each state transition is a sample from this model. Such a stochastic testing environment poses significant challenges for all algorithms. Notably, while our algorithms and Optimized use the same ensemble of world models, Optimized performs training directly on the ensemble without dynamically adapting the belief, leading to inferior performance due to the discrepancy between the learned and ground-truth world models. To ensure fair comparisons, we set a fixed number of training epochs for all algorithms. $\beta_n$ tracking is relatively easier (since its tracking target has only 1 dimension while the targets in Temperature and Rotation control have 4 dimensions), so we use fewer learning epochs for $\beta_n$ tracking.
>
> ### For Question 3:
>
> As noted in the first paragraph of Page 9, our implementation is based on Optimized (Lu et al., 2022), with minimal changes to its codebase and hyperparameter settings. Specifically, Optimized utilized Bayesian Optimization to determine key hyperparameters, including $\lambda$ and the size of the ensemble, and we inherited these hyperparameter setups. While further fine-tuning could potentially improve the performance of our algorithms, maintaining minimal changes in implementation ensures that the performance improvements are attributable to our algorithm design. A detailed hyperparameter setup is provided in Appendix C.
>
> In general, either Bayesian Optimization or grid search can be employed to determine these parameters. For example, $\lambda$ can be selected from several values within the range [0, 100], and the size of the ensemble can vary between 5 and 20.
>
> ### References:
>
> [1] Auger, David, Adrien Couetoux, and Olivier Teytaud. "Continuous upper confidence trees with polynomial exploration–consistency." ECML PKDD, 2013.
>
> [2] Niu, Yazhe, et al. "Lightzero: A unified benchmark for monte carlo tree search in general sequential decision scenarios." Neural Information Processing Systems, 2023.
>
> [3] Ye, Weirui, et al. "Mastering atari games with limited data." Neural Information Processing Systems, 2021.
>
> [4] Lu, Cong, et al. “Revisiting Design Choices in Offline Model Based Reinforcement Learning.” International Conference on Learning Representations, 2022.

---

> > ### Author Response · Authors · 2024-12-02
> >
> > Dear Reviewer grWz,
> >
> > Today marks the final day of the discussion phase. We would greatly appreciate it if you could review our responses and let us know if they have adequately addressed your concerns.
> >
> > Thank you for your time and consideration.

---

### Official Review · Reviewer_YGvg · 2024-11-03

**Soundness:** 3
**Presentation:** 2
**Contribution:** 2
**Rating:** 5
**Confidence:** 4

**Summary:**

This paper models offline model-based reinforcement learning as a Bayes Adaptive MDP, which enables the agent to leverage the model uncertainty during learning. By using MCTS with double progressive widening, the agent achieves good performance on the D4RL benchmark and three target-tracking tasks.

**Strengths:**

The paper investigates a promising direction of leveraging the Bayesian RL framework combined with MCTS in offline RL. The paper is well-motivated and the performance on the D4RL benchmark outperforms baselines.

**Weaknesses:**

Experiments:
- A more thorough analysis of the proposed method's performance is necessary. The observed performance improvement appears to result primarily from two factors: (1) the use of BAMDP, and (2) planning with MCTS. To verify this, it would be helpful to see an experiment where only BAMDP is applied, keeping other elements unchanged. While the BA-MBRL variant seems intended to isolate the effects of BAMDP, the paper does not clearly explain how this differs from other models like MOPO or MoReL. For instance, it’s unclear if the performance gap is solely due to BAMDP. Additionally, tracking the evolution of $b(\theta)$ during updates would strengthen the claims about the role of BAMDP.
- A comparison of runtime across methods would be valuable, especially to understand the impact of using BAMDP and MCTS on computational efficiency. Providing runtime metrics could help illustrate any trade-offs between performance gains and computational cost. (The paper only reports the runtime of Sampled EfficientZero.)
- The inclusion of Figure 1 is somewhat ambiguous. The authors seem to suggest that the proposed method outperforms Sampled EfficientZero. However, a more detailed analysis is needed. For example, what factors contribute to the performance difference with BA-MCTS-SL? Is the advantage primarily due to using the Bayesian RL framework?
-  In Line 472, the paper states, "both Sampled EfficientZero and BA-MCTS-SL rely on supervised learning... Thus, purely mimicking the search result may be less sample-efficient than policy gradient methods." However, in Line 406, "BA-MCTS-SL performs similarly to MA-MCTS, validating the effectiveness of both policy update mechanisms." This conclusion seems inconsistent.

Writing:
- Notation: what is hars'. It appears many times but I haven't found the definition. For example, in Algorithm1, the inputs of SIMULATE are defined as ($s, h$), $b(\theta)$, $d$ (Line 225), while in Line 231, the inputs of SIMULATE are $(s', hars'), b', d-1$.
- The update of $b(\theta)$ in Eq 4. should be clarifed.
- The figures should be added via PDF or SVG, otherwise, the figure can be blurry.

**Questions:**

See weaknesses above.

---

> ### Author Response · Authors · 2024-11-25
>
> ### For Weakness 1:
>
> In Table 1, the performance gap between BA-MBRL and Optimized is solely attributable to the use of BAMDP.
>
> As mentioned in the first paragraph of Section 5.1, BA-MBRL follows existing offline MBRL methods such as Optimized, MOPO, and MOReL, by learning an ensemble of world models, applying reward penalties to construct a pessimistic MDP, and using the pessimistic MDP as a surrogate simulator for RL training. Particularly, BA-MBRL and Optimized both use the standard deviation of the predictions across all ensemble members (i.e., Eq. (5)) as the reward penalty design. Thus, the **ONLY** difference between BA-MBRL and Optimized lies in whether the belief over each ensemble member (i.e., $b(\theta)$) is adapted according to Eq. (4). The benefit of this belief adaptation is clearly demonstrated in Table 1.
>
> In Appendix H of the revised manuscript, we show that employing a Bayes-adaptive ensemble, instead of a uniform ensemble, improves the prediction likelihood for the provided offline trajectories and reduces the prediction errors in imaginary rollouts. As illustrative examples of this performance improvement, Figure 4 tracks belief adaptations during an offline rollout and several imaginary rollouts.
>
> ### For Weakness 2:
>
> The use of MCTS provides both performance gains (as shown in Table 1 and Figure 2) and additional computational cost. However, this extra computation is incurred only during the offline training stage. To ensure real-time execution, MCTS is not used during deployment. Furthermore, leveraging parallel computation frameworks can significantly accelerate planning with MCTS (as in AlphaZero) and reduce the offline training time.
>
> Thank you for this suggestion. In Appendix E of the revised manuscript, we provide a detailed comparison of the computational costs of our proposed algorithms and Sampled EfficientZero.
>
> ### For Weakness 3:
>
> Sampled EfficientZero and BA-MCTS-SL both adopt MCTS and supervised-learning-based policy improvement. The performance difference between them could be attributed to several factors: (1) Sampled EfficientZero integrates model learning and policy training into a single stage, which significantly increases the learning difficulty (compared to BA-MCTS-SL). (2) Sampled EfficientZero relies on a single latent model as the world model and lacks a mechanism to address uncertainty in dynamics and reward predictions. In contrast, our algorithm learns an ensemble of world models, dynamically adapts beliefs for each ensemble member, and constructs a pessimistic MDP to mitigate over-estimation. (3) Sampled EfficientZero does not incorporate double progressive widening (Auger et al., 2013) in its planning process, which could enhance planning performance in environments with continuous state/action spaces and stochastic dynamics.
>
> ### For Weakness 4:
>
> As mentioned in Line 422, BA-MCTS-SL struggles on Walker2d, where a warm-up training phase (using BA-MBRL) is required to establish a better initial policy. Specifically, for Walker2d tasks, during the first 100 out of 500 training steps, we use BA-MBRL to warm up the policy before switching to BA-MCTS-SL for further improvement. Without this warm-up phase, the policy would fail. This observation supports the statement in Line 472: “purely mimicking the search result may be less sample-efficient than policy gradient methods, as it fails to account for the continuous nature of the action space.”
>
> ### For Weakness 5:
>
> Sorry for the confusion. In $(s, h)$, $h$ denotes the transition history ending at state $s$, which is a sequence of states, actions, and rewards. After taking an action $a$ at $s$, the agent receives a reward $r$ and transitions to the next state $s’$. The history is then updated to $h’=hars’$. This clarification has been added to the fourth paragraph of Section 4.2 in the revised manuscript.
>
> ### For Weakness 6:
>
> The update from $b(\theta)$ to $b’(\theta)$ in Eq. (4) follows the Bayesian rule: $b’(\theta)(i) \propto b(\theta)(i) P_{\theta}^{i}(s' | s, a) R_\theta^i(r | s, a)$, where $i=1, \cdots, K$ and $K$ is the size of the ensemble. $b’(\theta)$ and $b(\theta)$ represent the posterior and prior distributions, respectively, while $P_\theta^i(s' | s, a) R_\theta^i(r | s, a)$ is the likelihood of the transition $(s, a, r, s’)$ under ensemble member $i$ (i.e., ($P_\theta^i, R_\theta^i$)). We have clarified this in the second paragraph of Section 4.1.
>
> ### For Weakness 7:
>
> We are including the figures as EPS files and are happy to provide clearer versions of the images for the camera-ready version.
>
> ### References:
>
> [1] Auger, David, Adrien Couetoux, and Olivier Teytaud. "Continuous upper confidence trees with polynomial exploration–consistency." ECML PKDD, 2013.

---

> > ### Comment · Reviewer_YGvg · 2024-11-29
> >
> > Thank the authors for the explanation and the updated manuscript.
> > - In Line 403, the paper refers to BA-MCTS-SL as "replacing the policy learning algorithm in BA-MCTS from policy gradient methods (as in SAC) with supervised learning (SL)". And later in line 420, you mentioned that BA-MCTS-SL requires the warmup stage. From Table 1, it looks like the BA-MCTS-SL works better than the rest, is the warmup stage used in BA-MCTS-SL for Table 1 or only for the walker domain?
> > - For `h'=hars'`, does it means that you have a memory state `h`, which keeps updating once new state `s`, action `a`, and reward `r` are received? If so, I guess you might use h' $\leftarrow$ (h, a, r, s') as the equal relationship is not true, or you can use h' = $f$ (h, a, r, s'), where $f$ is the function used to update the state `h`. Please correct me if I am wrong.
> > - You can do this later, but for Fig 3 in the updated version, the plot seems a screenshot of the Sampled EfficientZero but your results are described in the caption.

---

> ### Author Response · Authors · 2024-11-29
>
> Thank you for your detailed feedback.
>
> - The warm-up stage was applied only to the Walker2d domain.
>
> - Yes, $h$ represents a memory state, updated as $h' = f(h, a, r, s')$, where $f$ denotes a concatenation/RNN operator. We will clarify this in the revised version.
>
> - Figure 3 consists of 12 subfigures, each rendered as an EPS file. We will refine this figure in the upcoming revision.

---

> > ### Comment · Reviewer_YGvg · 2024-11-29
> >
> > Thanks for the quick reply.
> >
> > I adjust my score accordingly. I think using the BAMDP is a nice idea and could be a good paper. I would suggest to keep polishing the paper. For example, changing the meaning of BA-MCTS-SL in the paper may cause unnecessary confusion. I think this clarity issue is mentioned by reviewers khHT, YGvg and grWz. The updated version improves the writing, but there is still room for improvement.

---

> > > ### Author Response · Authors · 2024-12-01
> > >
> > > Thank you. We will further improve the clarity in the next revision.

---

### Official Review · Reviewer_khHT · 2024-11-04

**Soundness:** 3
**Presentation:** 2
**Contribution:** 2
**Rating:** 6
**Confidence:** 4

**Summary:**

This paper addresses model uncertainty in offline MBRL by framing it as a BAMDP, allowing for adaptive belief updates over multiple potential world models. The authors propose a novel continuous MCTS method to solve BAMDPs in continuous and stochastic settings, integrating this planning approach as a policy improvement step in a policy iteration framework. The approach, which combines Bayesian RL with search, is shown to improve performance across a range of continuous control tasks.

**Strengths:**

* Practical MCTS implementation for offline MBRL formulated as BAMDP: the paper introduces a practical implementation of the MCTS algorithm specifically adapted for continuous control tasks under the BAMDP framework. Though a straightforward approach, this integration is effective and brings continuous MCTS to offline MBRL with clear potential.

* Well-executed ablative studies: the paper provides ablation studies that dissect the individual contributions of the algorithm’s components, demonstrating the benefits of the BAMDP formulation, MCTS planning, and the search-guided policy learning approach.

**Weaknesses:**

* Insufficient discussion of related work in Bayesian RL and offline MBRL: the related work section could better connect with existing research on modeling offline RL as a Bayes Adaptive MDP or epistemic POMDP. Ghosh et al. (2022) introduced a model-free approach to handling model uncertainty by training offline RL policies to adapt to belief changes, while Dorfman et al. (2021) framed offline meta-RL as a Bayesian RL problem, leveraging a belief-augmented MDP. Discussing these connections would position the current work more clearly within the broader landscape of Bayesian treatments of offline RL.

* Lack of connection to prior policy learning with model-based search: using model-based search outcomes for policy learning within an actor-critic framework was introduced by Feinberg et al. (2018) and later applied to offline MBRL in Jeong et al. (2023). Although these methods do not use MCTS for rollouts, they share the concept of leveraging generated trajectories to improve action-value estimation for policy learning. Discussing these earlier works would clarify the paper’s contributions in relation to prior approaches.

* Clarity issues in the experiment section: the structure of the experiment section could be improved to clarify the main research questions and key takeaways. At present, the content lacks focus on specific questions, making it challenging to follow the significance of each result. Organizing the discussion around clearly defined research questions could improve readability and focus.

* Omission of recent offline MBRL baselines: the empirical evaluations rely on older methods (e.g., MOPO and MOReL), making it difficult to assess claims about state-of-the-art performance. Comparisons with more recent baselines, such as RAMBO (Rigter et al., 2022), MAPLE (Chen et al., 2021), and CBOP (Jeong et al., 2023), would strengthen the experimental claims and provide a more comprehensive evaluation of the proposed method’s effectiveness.

__References:__

- Ghosh et al. (2022): "Offline RL policies should be trained to be adaptive."

- Dorfman et al. (2021): "Offline meta reinforcement learning – identifiability challenges and effective data collection strategies."

- Feinberg et al. (2018): "Model-Based Value Expansion for Efficient Model-Free Reinforcement Learning."

- Jeong et al. (2023): "Conservative Bayesian Model-Based Value Expansion for Offline Policy Optimization."

- Rigter et al. (2022): "Rambo-rl: Robust adversarial model-based offline reinforcement learning."

- Chen et al. (2021): "Offline model-based adaptable policy learning."

**Questions:**

* Belief updates during training: are the beliefs updated using imaginary rollouts sampled from the model ensemble? If so, could this introduce systematic biases? The paper mentions applying a reward penalty within a pessimistic MDP framework, but it would be useful to assess whether this is sufficient for accurate belief updates under the BAMDP setup.

* Representation of returned policy in Algorithm 1: how is the policy $\pi_{\mathrm{ret}}$, returned by the SEARCH procedure, represented in practice? Further details would clarify its applicability to real-time policy learning.

* Clarification on state representation: footnote 5 states that the policy takes the states consisting of both $s$ and $h$, but line 416 describes differently. Could the authors clarify this discrepancy? Additionally, MAPLE (Chen et al., 2021) employs an RNN-based policy network, which aligns well with the BAMDP framework. Using MAPLE as a baseline could enhance consistency with the proposed architecture.

---

> ### Author Response · Authors · 2024-11-25
>
> ### For Weakness 1 & 2:
>
> Thank you for your valuable feedback. We have revised the related work section of our paper, specifically the last paragraph of Section 3, to incorporate research on Bayesian approaches to offline RL and algorithms leveraging model-based search outcomes to enhance the efficiency of actor-critic training. We agree that including these prior works adds clarity to our contributions.
>
> ### For Weakness 3:
>
> We have improved the clarity of the experiment section (i.e., Section 5). Specifically, we restructured it into three subsections, clearly outlined the six main research questions (in the first paragraph of Section 5), and explicitly linked each result to the corresponding research question to emphasize their significance.
>
> ### For Weakness 4:
>
> As mentioned in the first paragraph of Page 9, our implementation is based on Optimized (Lu et al. (2022)), with minimal changes to its codebase and hyperparameter settings. In particular, the difference between BA-MBRL and Optimized is whether to adapt the belief over each ensemble member (i.e., $b(\theta)$) according to Eq. (4); the difference between BA-MBRL and BA-MCTS is whether to use MCTS to collect the training samples for downstream SAC; and the difference between BA-MCTS and BA-MCTS-SL is whether to supervised learning to train the policy function. **These strictly controlled experiments demonstrate that the performance improvements are driven by the Bayesian RL framework and the deep search component, which is our main experimental objective.** Further, both components can be seamlessly integrated with other advancements in offline MBRL, such as more accurate world model learning and improved uncertainty quantification for constructing pessimistic MDPs.
>
> In Appendix I of the revised manuscript, we provide comparisons with RAMBO (Rigter et al., 2022), MAPLE (Chen et al., 2021), and CBOP (Jeong et al., 2023), demonstrating that our algorithm achieves state-of-the-art performance.

---

> ### Author Response · Authors · 2024-11-25
>
> ### For Question 1:
>
> To mitigate error accumulation, we did not do full trajectory rollouts. As mentioned in the first paragraph of Appendix C, at each learning epoch, 50000 states are randomly sampled from the offline dataset, with each state followed by a rollout lasting $H$ time steps. $H$ is no more than 50, according to Table 3. Suppose a state $s_t$ is sampled from the offline dataset. The prior distribution $b_t(\theta)$ at $s_t$ is computed as follows. The prior distribution at the initial state $s_0$ of the trajectory containing $s_t$ is a uniform distribution $b_0(\theta)$. This distribution is then adapted along the offline trajectory, according to Eq. (4), until $b_t(\theta)$ is obtained. Consequently, most belief adaptations are based on the offline trajectories (which are collected from the true environment), with additional adaptations conducted during imaginary rollouts using the learned models. We have updated Algorithm 2 and Section 4.3, with the modifications highlighted in blue, to clarify this issue.
>
> Systematic biases may occur in regions with low data coverage. To address this, we construct a pessimistic MDP to discourage the agent from exploring these regions or overestimating the planning results within them. Also, in Appendix H, we show that employing a Bayes-adaptive ensemble, instead of a uniform ensemble, significantly reduces the prediction errors in imaginary rollouts
>
> ### For Question 2:
>
> We provided a discussion on the design choices of $\pi_{\text{ret}}$ in the third paragraph of Appendix B. Specifically, $\pi_{\text{ret}}(a|(s, h)) \propto N((s, h), a)^{1/\tau},\ a \in C((s, h))$. Here, $(s, h)$ is a decision node, $C((s, h))$ is the set of sampled actions at $(s, h)$, $N((s, h), a)$ is the number of times $a$ has been sampled at $(s, h)$, $\tau > 0$ is the temperature parameter which decreases with the training process. This design follows MuZero and is practical since it’s defined based on a finite set of actions sampled at $(s, h)$.
>
> ### For Question 3:
>
> As noted in the first paragraph of Page 9, for fair comparisons, we adopt the same policy architecture as the baselines, i.e., a feedforward neural network, rather than an RNN that incorporates transition history as input. We agree that involving the transition history $h$ as part of the input and adopting an RNN as the policy network could further improve the empirical performance. However, we choose to apply minimal changes to the codebase of existing offline MBRL algorithms like Optimized. This ensures that the performance improvements reported in Table 1 are attributable to dynamically adapting the belief over ensemble members or using MCTS for enhanced planning.
>
> Although $h$ is not directly used as an input to $\pi$, it is utilized to update the belief distribution $b(\theta)$, which influences both the acting and planning phases. The policy $\pi(a|s)$ is trained based on the planning results and can therefore be viewed as a distilled function of $\pi(a|s, h)$. In this sense, the policy implicitly accounts for $h$ through belief adaptations.
>
> In Appendix I, we demonstrate that BA-MCTS outperforms MAPLE (Chen et al., 2021) on the D4RL MuJoCo benchmark.
>
> ### References:
>
> [1] Lu et al. (2022):  "Revisiting design choices in offline model based reinforcement learning." ICLR.
>
> [2] Rigter et al. (2022): "Rambo-rl: Robust adversarial model-based offline reinforcement learning." NeurIPS.
>
> [3] Chen et al. (2021): "Offline model-based adaptable policy learning." NeurIPS.
>
> [4] Jeong et al. (2023): "Conservative Bayesian Model-Based Value Expansion for Offline Policy Optimization." ICLR.

---

> > ### Author Response · Authors · 2024-12-02
> >
> > Dear Reviewer khHT,
> >
> > Today marks the final day of the discussion phase. We would greatly appreciate it if you could review our responses and let us know if they have adequately addressed your concerns.
> >
> > Thank you for your time and consideration.

---

> > > ### Comment · Reviewer_khHT · 2024-12-03
> > >
> > > Thank you for the responses. I have updated my score to 6 based on the rebuttal responses and updated paper.

---

> > > > ### Author Response · Authors · 2024-12-03
> > > >
> > > > Thank you for your thoughtful evaluation and the updated score.

---

### Official Review · Reviewer_HEZk · 2024-11-05

**Soundness:** 2
**Presentation:** 2
**Contribution:** 2
**Rating:** 5
**Confidence:** 4

**Summary:**

This paper propose to use a Bayesian adaptive method to train the model instead of end to end training approach used by MuZero. As indicate by the authors, this can modify the difficulty of model training in complex scenarios like continuous action space. Specifically, the method can be described in three main designs:1) it adapts DPW to handle continuous action and state spaces. 2) it use BA to refine the model during interactions and search process 3) It adapts MuZero’s supervised way to perform policy improvement. The experiment section shows methods using BA generally have a better performance than baselines, but the advantage of certain design seems not significant.

**Strengths:**

1. The authors describe the entire method in a clear way using pseudocode in Algorithms 1 and 2, and the insight of separating the model training from end-to-end process is explained in experiment section by comparing to sampled EfficientZero (sez).
2. The experimental results indicate the effectiveness of the BA model training. Although the effectiveness of certain designs like MCTS search and supervised policy learning is not significant according to Table 1, I am glad to see the authors report the results honestly.

**Weaknesses:**

1. It seems the main improvement comes from BA model training according to Table 1. Thus, I wonder if there are other model-based RL methods using BA and deep ensembles to train the models, and why the authors did not include them in the baseline.
2. In my opinion, the BA process adjust the belief of the models according to the sample results of current belief instead of the interaction with true environment, which may lead to accumulated error. For example, if the belief mistakenly concentrates on a wrong model and the sample results are likely to come from the wrong model, then future belief are likely to be more inclined to this wrong model. Could the authors provide some insights that why this not happen according to the experiment results?

**Questions:**

1. I am not sure about the necessity of adjusting the belief not only in the acting process but also in the searching process. Could the author explain the different influences of adjusting the model at the two stages?
2. The authors indicate that sez does not work well due to the difficulty of end to end model training, but sez also use a different way of choosing actions nodes from DPW. Can the authors explain why they do not consider it as a main factor?
3. I am curious about why the authors choose to update the policy in a trajectory way. MuZero just sample a series of transitions from the buffer, which can avoid the interaction in learned models. Could the authors explain the reason behind this design?

---

> ### Author Response · Authors · 2024-11-25
>
> ### Regarding Weakness 1:
>
> To the best of our knowledge, there is no other offline model-based RL algorithm using BAMDP.
>
> Also, we’d like to point out that the improvements stem from both the use of BAMDP and the incorporation of MCTS. In Table 1, the comparison between BA-MBRL and Optimized highlights the benefit of using BAMDP, as it is the only difference in their algorithm designs. Further, compared to BA-MBRL, BA-MCTS introduces a planning process with MCTS, which results in significant performance improvements, as shown in Table 1.
>
> ### Regarding Weakness 2:
>
> In the offline learning setting, we don’t have access to the true world model, so we can only generate rollouts based on learned world models. To mitigate error accumulation, we did not do full trajectory rollouts. As mentioned in the first paragraph of Appendix C, at each learning epoch, 50000 states are randomly sampled from the offline dataset, with each state followed by a rollout lasting $H$ time steps. $H$ is no more than 50, according to Table 3. Suppose a state $s_t$ is sampled from the offline dataset. The prior distribution $b_t(\theta)$ at $s_t$ is computed as follows. The prior distribution at the initial state $s_0$ of the trajectory containing $s_t$ is a uniform distribution $b_0(\theta)$. This distribution is then adapted along the offline trajectory, according to Eq. (4), until $b_t(\theta)$ is obtained. Consequently, most belief adaptations are based on the offline trajectories (which are collected from the true environment), with additional adaptations conducted during imaginary rollouts using the learned models. We have updated Algorithm 2 and Section 4.3, with the modifications highlighted in blue, to clarify this issue.
>
> For your example, we think it’s less likely that the belief would mistakenly concentrate on a wrong model. According to Section 4.1, the belief starts with a uniform distribution and its update is based on either the offline trajectories or imaginary rollouts generated by the predictions of all ensemble members. Thus, for a belief to concentrate on a wrong model, it needs more ensemble members to agree on a wrong prediction than a correct prediction, which is less likely to happen in regions with good data coverage. While, for predictions in out-of-sample regions, it could be problematic but, as mentioned in the last paragraph of Section 4.1, we construct a pessimistic MDP to discourage the agent from exploring those regions or overestimating the planning results there.

---

> > ### Comment · Reviewer_HEZk · 2024-12-03
> > **Discussion**
> >
> > **For Weakness 1:**
> >
> > The first offline model-based RL algorithm utilizing BAMDPs is indeed a significant contribution. However, the authors need to provide better comparative forms and visualizations to illustrate the significance of combining BAMDP with MCTS. It is crucial to articulate the innovation of this integration more explicitly, rather than presenting it as a combination of A and B. Additionally, a substantial portion of the experiments in this paper are conducted on the D4RL locomotion tasks. Given the specific definitions of these environments, they do not necessitate overly complex planning; mastering basic control operations suffices. Within a single episode, there is minimal variation in behavior.
> >
> > **For Weakness 2:**
> >
> > The authors cannot solely rely on textual descriptions or final performance results to assert the absence of accumulated errors. It would be beneficial to include visualizations of the offline dataset to demonstrate the ability to fully learn the belief and exhibit interpolation and extrapolation capabilities. Nevertheless, the use of pessimistic MDP-related designs is a commendable approach that can mitigate this issue to some extent.
> >
> > **For Q1:**
> >
> > I understand the authors' motivation to maintain overall consistency, and the integration of ensemble techniques with Bayesian RL is indeed well-designed. However, a detail to note is that Appendix C reveals the authors have fine-tuned hyperparameters (specifically, ensemble size) for different environments. Perhaps the authors could further refine the design of the ensemble technique, for example, drawing inspiration from the EDAC paper in the model-free offline RL domain.
> >
> > **For Q2:**
> >
> > I fully agree with the authors' analysis of addressing uncertainty in dynamics and reward predictions. However, the two-stage approach may not be strictly necessary; uncertainty-related control and trade-offs could potentially be integrated into an end-to-end solution. Additionally, I believe the experiments on stochastic environments should be placed earlier in the paper, as these types of problems better highlight the advantages of the proposed method. Lastly, prediction errors in deep networks during RL training can also be considered a form of randomness, suggesting that your method could potentially be effective from this perspective as well.
> >
> >
> > In summary, while the authors have addressed some of my questions, some improvements are still required for this paper to meet the acceptance criteria. Therefore, I maintain my original score.

---

> > > ### Author Response · Authors · 2024-12-03
> > >
> > > Thank you for taking the time to read our response and for providing detailed feedback. Below, we address the points you raised:
> > >
> > > ### For Weakness 1:
> > >
> > > This paper does not aim to explore the combination of BAMDP and MCTS. Instead, we propose modeling offline model-based RL as solving a BAMDP (Section 4.1). Based on prior literature, BAMCP (a planner built on MCTS [1]) is commonly used to solve BAMDPs. However, BAMCP cannot handle continuous control problems. To address this limitation, we propose Continuous BAMCP as an extension in Section 4.2. Importantly, our approach is flexible: any scalable planner for BAMDPs, including model-free RL methods, could replace Continuous BAMCP. This flexibility is why we compare BA-MCTS with BA-MBRL in Section 5 to highlight the benefit of deep search.
> > >
> > > In Section 5.3, we provide evaluations on challenging Tokamak control tasks, which require complex planning and precise control, further validating our approach.
> > >
> > > ### For Weakness 2:
> > >
> > > As shown in Tables 9 and 10 of Appendix H, incorporating belief adaptation helps mitigate accumulated errors in both replaying offline trajectories and performing imaginary rollouts using the learned models. These results highlight the interpolation and extrapolation capabilities of belief adaptation, respectively.
> > >
> > > Including visualizations of the offline dataset to demonstrate the ability to fully learn the belief is impractical because beliefs are numerical results and the ground truth for beliefs over each ensemble member is inaccessible. Instead, we indirectly demonstrate the benefits of belief adaptation through the reduction in prediction errors, as reflected in Tables 9 and 10.
> > >
> > >  ### For Q1:
> > >
> > > As discussed in [2], ensemble size is a critical hyperparameter in offline MBRL. In our work, we adopt the same hyperparameter setup as theirs.
> > >
> > > Thank you for your suggestion. As noted in the first paragraph of Page 9, both components of our approach (i.e., Bayesian RL and deep search) can be integrated with other advancements in offline MBRL, such as improved world model learning (e.g., through refined ensemble techniques).
> > >
> > > ### For Q2:
> > >
> > > Thank you for your valuable advice. We will present the experiments on stochastic environments earlier in the paper. An end-to-end solution, which integrates Bayesian RL, deep search, and offline MBRL, is indeed an exciting future direction for exploration.
> > >
> > > ### References:
> > >
> > > [1] Guez, Arthur, David Silver, and Peter Dayan. "Scalable and efficient Bayes-adaptive reinforcement learning based on Monte-Carlo tree search." Journal of Artificial Intelligence Research 48 (2013): 841-883.
> > >
> > > [2] Lu, Cong, et al. “Revisiting Design Choices in Offline Model Based Reinforcement Learning.” International Conference on Learning Representations, 2022.

---

> ### Author Response · Authors · 2024-11-25
>
> ### For Question 1:
>
> The purpose of search/planning is to simulate the acting process by looking ahead. Therefore, the search and acting processes should be based on the same transition dynamics, that is, utilizing the same ensemble and the same Bayesian adaptive process.
>
> As discussed in Section 4.2, the search algorithm is designed to address a planning problem within a BAMDP (defined in Section 2). According to Eqs. (1) and (4), state and reward transitions should be based on predictions from all ensemble members, with the belief updated at each time step based on the transition outcomes.
>
> ### For Question 2:
>
> Thank you for the question. We need to adjust our analysis of the performance difference between Sampled EfficientZero and BA-MCTS-SL as follows:
>
> Sampled EfficientZero and BA-MCTS-SL both adopt MCTS and supervised-learning-based policy improvement. The performance difference between them could be attributed to several factors: (1) Sampled EfficientZero integrates model learning and policy training into a single stage, which significantly increases the learning difficulty (compared to BA-MCTS-SL). (2) Sampled EfficientZero relies on a single latent model as the world model and lacks a mechanism to address uncertainty in dynamics and reward predictions. In contrast, our algorithm learns an ensemble of world models, dynamically adapts beliefs for each ensemble member, and constructs a pessimistic MDP to mitigate over-estimation. (3) Sampled EfficientZero does not incorporate double progressive widening (Auger et al., 2013) in its planning process, which could enhance planning performance in environments with continuous state/action spaces and stochastic dynamics.
>
> ### For Question 3:
>
> As mentioned in Appendix C, at each learning epoch, $50000H$ transitions are sampled by
> interacting with the learned models, followed by 1000 RL training iterations. In particular, 50000 states are randomly sampled from the offline dataset, with each state followed by a rollout lasting $H$ time steps. Thus, the training is based on multiple imaginary trajectory segments rather than full trajectories. (Modifications have been made in Section 4.3 for clarification.) To mitigate overestimation caused by model inaccuracies, a reward penalty based on the discrepancy among ensemble members is applied, as defined in Eq. (5). This training protocol is adapted from Optimized and MOPO, which serve as our baselines. We follow the same process to ensure fairness in the evaluation.
>
> Offline variants of MuZero (Schrittwieser et al., 2021) train the policy network exclusively on states from the offline dataset and the planning results derived from those states. This training approach can be somewhat restrictive, particularly when the offline dataset fails to provide adequate coverage of the continuous state space.
>
> ### References:
>
> [1] Auger, David, Adrien Couetoux, and Olivier Teytaud. "Continuous upper confidence trees with polynomial exploration–consistency." ECML PKDD, 2013.
>
> [2] Schrittwieser, Julian, et al. "Online and offline reinforcement learning by planning with a learned model." Neural Information Processing Systems, 2021.

---

> > ### Author Response · Authors · 2024-12-02
> >
> > Dear Reviewer HEZk,
> >
> > Today marks the final day of the discussion phase. We would greatly appreciate it if you could review our responses and let us know if they have adequately addressed your concerns.
> >
> > Thank you for your time and consideration.

---

### Author Response · Authors · 2024-11-25
**Revision Details**

Thanks to the valuable feedback from all reviewers, we have made significant improvements in this revision. All modifications are highlighted in blue for ease of reference.

- **Related Work (Section 3)**: We restructured this section by adding paragraph titles and providing a more structured comparison with key related works, including MuZero, Bayesian approaches to offline RL, and algorithms that leverage model-based search to enhance actor-critic training efficiency. Incorporating these prior works clarifies our contributions.

- **Belief Adaptation**: We added explanations for the belief adaptation process (i.e., Eq. (4)) and clarified notations such as $hars'$.

- **Section 4.2**: This section was refined by reorganizing the content and adding paragraph titles. We also clarified the design motivations and emphasized our novel contributions.

- **Training Framework (Section 4.3)**: We modified the overall training framework, including the pseudocode (Alg. 2) and the second paragraph in Section 4.3, to align with the experimental setup described in Appendix C. Additionally, we clarified that training is based on imaginary trajectory segments, rather than full trajectories, consistent with standard offline MBRL methods like Optimized and MOPO.

- **Experiment Section (Section 5)**: We improved the clarity of this section by restructuring it into three subsections, explicitly outlining six main research questions (in the first paragraph), and linking each result to its corresponding research question to highlight their significance.

- **Appendices**:
  - **Appendix E**: Reports the computational cost of our algorithms compared to Sampled EfficientZero.
  - **Appendix F**: Includes an ablation study showing the necessity of using a reward penalty to construct a pessimistic MDP.
  - **Appendix H**: Adds two tables and four figures to demonstrate the benefits of belief adaptation. Using a Bayes-adaptive ensemble, instead of a uniform ensemble, improves prediction likelihood for offline trajectories and reduces prediction errors in imaginary rollouts. Figure 4 illustrates these improvements by tracking belief adaptations during an offline rollout and several imaginary rollouts.
  - **Appendix I**: Adds a table comparing our algorithm with additional baselines, including MOBILE (ICML 2023), CBOP (ICLR 2023), RAMBO (NeurIPS 2022), APE-V (ICML 2022), and MAPLE (NeurIPS 2021), demonstrating our state-of-the-art performance.

To summarize, we significantly improved the clarity of the paper and added three new sets of experiments. The codes to reproduce these results are included in the supplementary materials.

---

### Meta-Review · Area_Chair_oYQT · 2024-12-21

**Metareview:**

This paper propose to use Bayesian adaptive method for offline RL. Main drawback of the paper include: (1) a lack of theoretical justification of why a Bayesian framework helps with mitigating the out-of-distribution generalization problem in offline RL, and how one should pick the prior in practice. (2) the clarity of the paper can be improved. (3) in the experimental evaluation, more modern and SOTA baseline should be included.

**Additional Comments On Reviewer Discussion:**

NA

---

### Decision · Program_Chairs · 2025-01-22

Reject